# T1: A Tool-Oriented Conversational Dataset for Multi-Turn Agentic Planning

**Amartya Chakraborty**\*, **Paresh Dashore**\*, **Nadia Bathaee**\*, **Anmol Jain**\*,
**Anirban Das**, **Shi-Xiong Zhang**, **Sambit Sahu**, **Milind Naphade**, **Genta Indra Winata**\*
Capital One
{amartya.chakraborty, paresh.dashore, nadia.bathaee}@capitalone.com
{anmol.jain, genta.winata}@capitalone.com

## Abstract

Large Language Models (LLMs) have demonstrated impressive capabilities as intelligent agents capable of solving complex problems. However, effective planning in scenarios involving dependencies between API or tool calls-particularly in multi-turn conversations-remains a significant challenge. To address this, we introduce T1, a tool-augmented, multi-domain, multi-turn conversational dataset specifically designed to capture and manage inter-tool dependencies across diverse domains. T1 enables rigorous evaluation of agents' ability to coordinate tool use across nine distinct domains (4 single domain and 5 multi-domain) with the help of an integrated caching mechanism for both short- and long-term memory, while supporting dynamic replanning-such as deciding whether to recompute or reuse cached results. Beyond facilitating research on tool use and planning, T1 also serves as a benchmark for evaluating the performance of open-weight and proprietary large language models. We present results powered by T1-AGENT, highlighting their ability to plan and reason in complex, tool-dependent scenarios.

## 1   Introduction

Leveraging external tools through Large Language Models (LLMs) to solve diverse conversational tasks has emerged as a promising direction in the development of intelligent agents [15]. Despite recent progress, task-oriented LLM-based dialogue systems continue to struggle with long conversational contexts [3]. Moreover, there remains a lack of comprehensive resources for training and evaluating multi-turn, multi-domain conversational agents that address complex user needs. Existing datasets primarily focus on single-turn interactions for planning tasks, such as executing APIs or code [12, 20], and fail to capture realistic multi-turn scenarios that require reasoning over extended contexts, coordination across multiple tools, and adherence to intricate constraints. In real-world settings, tool use often involves interdependent tools, where the correctness and efficiency of task completion rely heavily on the context and the order in which tools are invoked. An effective agent must therefore be capable of understanding when to call a tool, which tool to use, and in what sequence, in order to successfully complete complex tasks.

To address this gap, we introduce T1, a new dataset and evaluation framework for assessing agent performance in multi-turn dialogues, with a particular focus on tool usage and reasoning about inter-tool dependencies. The dataset spans multiple domains and features complex, goal-oriented interactions between users and a travel assistant. Alongside the dataset, we propose T1-AGENT, an agent designed to interpret nuanced user intents and generate executable code using a predefined set of tools. T1 is specifically designed to evaluate the ability of LLM-based agents to plan tool use

---

*The authors contributed equally. We publicly release our code and dataset at `https://github.com/CapitalOne-Research/T1` and `https://huggingface.co/datasets/capitalone/T1`, respectively.

39th Conference on Neural Information Processing Systems (NeurIPS 2025) Track on Datasets and Benchmarks.

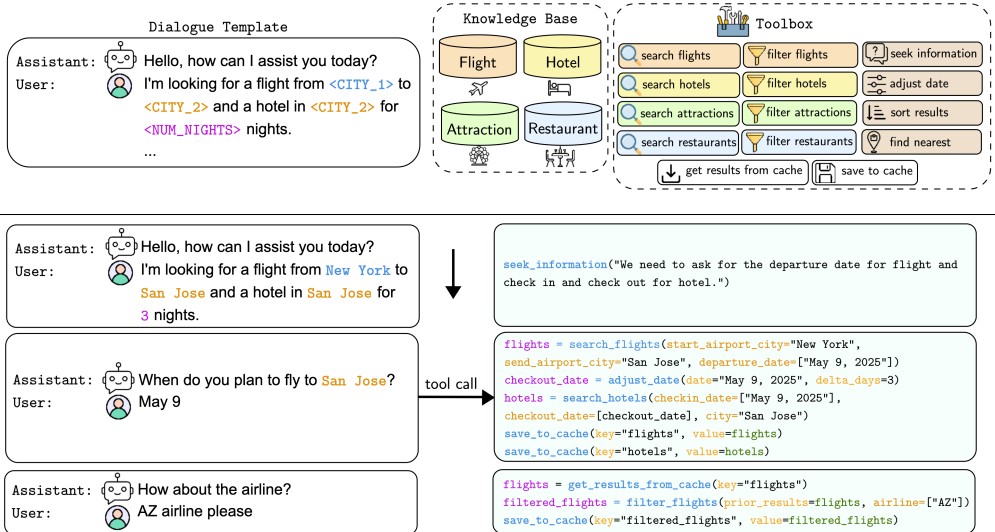

Figure 1: Illustrative example from the T1 dataset. This example showcases a multi-domain scenario involving both flights and hotels, where the user is planning a trip and attempting to book relevant services. The dialogue is constructed by retrieving entities from a knowledge base, and tool calls are executed using a predefined toolbox, simulating realistic, tool-augmented agent behavior.

effectively and leverage a caching mechanism to efficiently reuse previously retrieved information. To support this, we incorporate dedicated tools for accessing and managing the cache.

Our contributions can be summarized as follows:

- We introduce T1, a comprehensive multi-turn dataset consisting of 13.5k dialogues designed to evaluate tool-using, LLM-based agents across nine key domains—comprising four single-domain and five multi-domain settings. The dataset covers a wide range of interaction scenarios, including single-domain, mixed-domain, and fully multi-domain conversations. It incorporates 14 distinct tools, enabling realistic and fine-grained assessment of agent capabilities in complex, tool-driven dialogue tasks.

- To enhance the complexity and realism of the evaluation, the dataset includes cross-domain tasks and interdependent tool calls, requiring agents to reason about tool selection and execution order within context. This evaluation framework assesses the ability of LLM-based agents to think critically, reason effectively, and make context-aware decisions.

- We evaluate our dataset using an LLM-based agent, T1-AGENT, a code-generation system built on open-weight language models and equipped with a caching mechanism for improved performance. This architecture enables scalable evaluation and provides a robust, efficient framework for tool-using agents.

## 2 T1 Dataset

T1 is a dataset specifically designed to evaluate LLM-based agents on tool usage and complex planning tasks over multi-turn conversational context. This dataset simulates multi-turn conversations spanning both four single-domain and five multi-domain settings: *flights*, *restaurants*, *hotels*, *attractions*, *flights-hotels*, *hotels-restaurants*, *hotels-attractions*, *flights-hotels-attractions*, and *flights-hotels-restaurants*. Planning tasks are formulated as code, where function calls to external tools are used to accomplish specific goals.

### 2.1 Tasks and Notations

We define a dialogue $D$ as an alternating sequence of assistant and user turns:

$$D = \{A_1, U_1, A_2, U_2, \ldots, A_n\}, \tag{1}$$

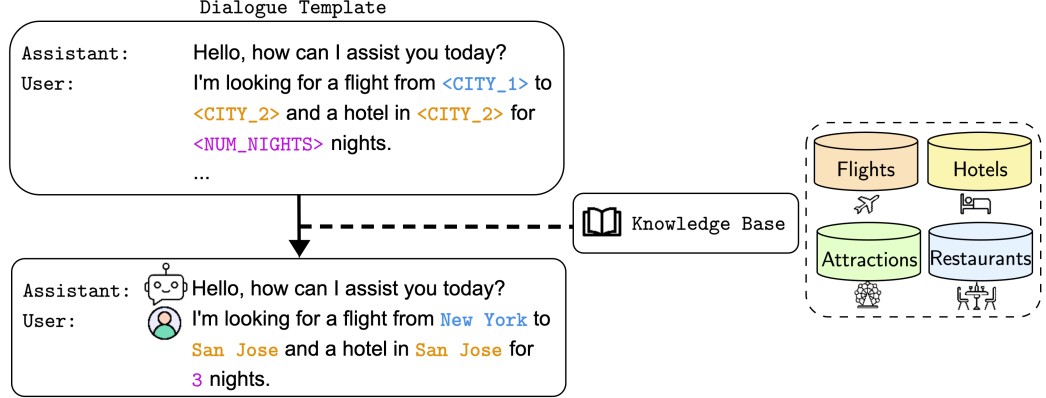

Figure 2: T1 generates data by populating delexicalized entities with corresponding entries from the knowledge base.

where each $A_i$ represents an assistant turn and $U_i$ a user turn. The dialogue always starts with an assistant turn and proceeds in a strictly alternating order. We provide a set of tools $\mathcal{T}$, where each tool $t \in \mathcal{T}$ encapsulates logic to perform a specific function. These tools can be categorized as follows:

- **Domain-specific tools**: Designed to handle operations tied to a particular application domain.

- **Interdependent tools**: Used to identify or reason about dependencies across domains.

- **Generic tools**: Domain-independent or auxiliary utilities applicable across tasks.

## 2.2 Dataset Construction

We construct our dataset by manually collecting data from Wikipedia to gather entities and metadata. For this task, we define four domains: flights, hotels, restaurants, and attractions. Additionally, we compile a list of 128 airports and 321 cities within the United States, along with up to 15 neighborhoods for each city. Some information—such as airline names, hotel names, and hotel star ratings—is synthesized to avoid inaccuracies and to prevent the LLMs from relying on its internal knowledge of real-world named entities.

| Dataset | Deployed Tools | Human Annotated Tool Planning | Execution Result Evaluation | Multi-turn Context Planning | Multi-Domain Tool Planning |
|---|:---:|:---:|:---:|:---:|:---:|
| APIBank [4] | ✓ | ✓ | ✓ | ✓ | |
| APIBench [8] | | | ✓ | | |
| GAIA [6] | | ✓ | ✓ | | |
| GTA [18] | ✓ | ✓ | ✓ | ✓ | |
| m&m's [5] | ✓ | ✓ | ✓ | ✓ | |
| ToolBench [11] | ✓ | | ✓ | | |
| Toolformer [12] | ✓ | ✓ | ✓ | | |
| TravelPlanner [20] | ✓ | ✓ | ✓ | | ✓ |
| **T1** | ✓ | ✓ | ✓ | ✓ | ✓ |

Table 1: Comparison of datasets for the LLM-based agent systems.

### 2.2.1 Ontology

We have a total of 5 ontologies, one for each of the four defined domains and another one for city. For each ontology, we curated a list of key attributes that would be relevant such as the airline or number of layovers for a flight, the cuisine of a restaurant or the number of stars or customer rating for a hotel. For each of these attributes, we defined the possible values and then used the ontology to generate synthetic data for flights, hotels and restaurants which would subsequently be used by the tools we defined.

| | Single-domain | | | | Multi-domain | | Common | Total |
| | Flights | Hotels | Restaurants | Attractions | 2 Domains | 3 Domains | | |
|---|---|---|---|---|---|---|---|---|
| # Attributes | 35 | 43 | 21 | 7 | N/A | N/A | N/A | 106 |
| # Tools | 2 | 2 | 2 | 2 | 1 | N/A | 5 | 14 |
| # Dialogues | 1.5k | 1.5k | 1.5k | 1.5k | 4.5k | 3k | N/A | 13.5k |
| Avg. # turns | 8.2 | 8.0 | 9.0 | 6.0 | 11.3 | 9.7 | N/A | N/A |

Table 2: Dataset statistics detailing the number of attributes, tools, dialogues, and the average number of turns, categorized by each individual domain, multi-domain, and common categories. "N/A" indicates that the metric is not applicable in the given context.

**Flights.** The ontology includes the airline, flight class (`economy, business and first`), number of layovers ranging between `0-2 stops`, the duration of a layover ranging from `1-6 hours` and a list of the possible airports to depart from and arrive to. To see the full table, go to Table 37.

**Hotels.** The ontology includes the number of stars ranging between 1 to 5 stars, the customer rating of the hotel ranging from 1.0 to 5.0, the cost of the hotel as well as whether or not the hotel includes a number of amenities such as the presence of a gym or pool. To see the full table, go to Table 38.

**Restaurants.** The ontology includes the type of cuisine, the customer rating of the hotel ranging from 1.0 to 5.0, the price per person and whether or not the restaurant served any particular dietary options such as vegetarian, vegan, or halal. To see a full list of dietary restrictions, taken into consideration, go to Table 39.

**Attractions.** The ontology includes the type of attraction which is one of the following: touristy, culinary, historical, scenic, social, art, cultural, guided, and sporting.

**Cities.** The ontology includes a list of cities in the United States of America (US) that was collected through Wikipedia. For each city, we then extracted up to 15 neighborhoods as well as the approximate geographical coordinates of each neighborhood.

### 2.2.2 Knowledge Bases

**Attractions.** From the list of 321 cities, we collect up to 15 attractions for 85 cities through the usage of `Llama-3.3 70B Instruct` and conduct quality assurance by human annotators to ensure the data correctness. We also collect the city neighborhood for each attraction as well as the geographical coordinates. In total, there are 728 attractions that were collected from the 85 cities for this dataset.

**Flights.** The flight dataset includes a total of 128 airports used for data generation. Each flight is associated with an airline, randomly selected from the provided ontology. The departure and arrival airports are also randomly chosen from the list of airports in the ontology. While the departure time is randomly generated, the arrival time is calculated based on the geographical distance between the two airports, assuming an average flight speed of 450 miles per hour. In total, 480,410 synthetic flights were generated for this dataset.

**Hotels.** Hotels are generated for all 321 cities in the ontology. For a particular city, a neighborhood is assigned and additionally, synthetic latitude and longitude coordinates are generated for each hotel in a city as the coordinates would be used for distance computing purposes. Each hotel is also provided a star and a synthetic customer rating that would be correlated to the amenities offered by the establishment as well as the price per night. 47,589 hotels were generated as part of this dataset.

**Restaurants.** Restaurants are generated for all 321 cities in the ontology. Just like for hotels, in a particular city, a neighborhood is assigned to a restaurant and additionally, synthetic latitude and longitude coordinates are generated. Each restaurant also is given a user rating which was synthetically generated, a cuisine provided by the ontology as well as whether particular dietary options are supported and the average cost per person. 17,975 restaurants are generated as part of this dataset.

### 2.2.3 Data Annotation and Quality Assurance

To ensure high-quality and natural data, we employ five human annotators, with each data sample reviewed by both an annotator and a quality assurance (QA) reviewer. The annotators were selected to represent a diverse set of background and perspectives, while maintaining a high technical bar. All annotators have at least a Master's degree in Computer Science and demonstrated proficiency in Python, enabling them to handle complex annotation tasks requiring logical reasoning and scripting. The QA specialist also has a strong background in programming. Annotators are assigned a category of templates and are responsible for writing the appropriate code using the tools defined for this project. Afterward, the QA reviewer evaluates the annotated code and provides feedback, which is used to make necessary corrections and improvements.

### 2.2.4 Dialogue Generation

We create a total of nine data categories, as discussed in Section 4.1. The data construction follows a three-step process: first, we generate templates with placeholder values; second, we annotate the templates with code using the provided tools; and third, we programmatically fill in the placeholder values for each template.

**Generating Dialogue Templates.** For each of the 9 dialogue categories, we generate 60 dialogue templates using `Llama-3.3 70B Instruct`. These templates are then reviewed and refined by human annotators to ensure accurate dialogue flow, as well as high coherence and fluency. The model is prompted to generate synthetic dialogues for both single and multi-domain. Each template consists of a conversation between the assistant and user. To learn more about the prompt used, go to Appendix H and I. Additionally, the templates consist of placeholders for attributes such as a city or neighborhood name, cuisine of a restaurant, rating of a hotel or the type of an attraction. As the dataset is related to the area of travel, some placeholder values are tied to a particular city such as the neighborhood (`<CITY_x_NEIGHBORHOOD_x>`), airport name (`<CITY_x_AIRPORT_x>`), hotel name (`<CITY_x_HOTEL_NAME_x>`) and restaurant name (`<CITY_x_RESTAURANT_NAME_x>`).

**Template Lexicalization.** Within each dialogue category, 25 dialogues are generated with the placeholder values filled with the actual values present within the ontology. In addition to the dialogues, the ground-truth code which is part of the annotation has the placeholder values replaced with actual values. The dataset is split into three partitions: training, validation and test. To limit data contamination among these three partitions, the cities that are to be used for filling in the placeholder are also assigned to a particular partition only to be used there. Hence, for example if the city of Boston is assigned to the training partition, it will never be present in any of the validation or test dialogue conversations. Additionally, for fields such as the check-in and checkout dates for hotels or departure and return flight dates, particular methods are taken to ensure that any dates or times follow chronological order and there is no instance where a check-in date at a hotel would be after the checkout date.

**Generated Dialogue and Ground Truth Validation.** Once the placeholders are filled for both the dialogues and ground truth code samples, each block of code is executed to identify any potential errors in annotation including an improper use of a placeholder or a syntax error. The validation assists the team in correcting any annotations and ensures that the resulting code is runnable.

## 3 T1-Agent

We build an LLM powered T1-AGENT to evaluate and simulate our T1 agentic dataset and measure its performance across three tasks: information seeking, parameter extraction, and tool calling.

### 3.1 Information Seeking

Each tool has a mandatory set of parameters that must be provided before running the tool successfully. Information seeking is the task of gathering this mandatory set of parameters for any of the respective tools. We want to evaluate the capability of the agent to understand both the intent of the user's query and which parameters to ask the user about as a follow-up. Figure 1 shows an example of how information seeking is used by the agent. When the user inquires about a flight from New York to

San Jose and a hotel in San Jose, the agent understands that the user would like to search for flights but needs to provide the departure date of the flight and the check-in date of the hotel. Thus, the agent infers that in followup discussion, it needs to ask for these details from the user.

## 3.2 Parameter Extraction

After the agent understands the tools to call and the necessary parameters to collect, it works to extract those parameters from the user dialogue. From Figure 1, when the user mentions that they want a flight from New York to San Jose for 3 nights, the agent is able to extract the starting and ending airport city for the flight as well as the number of nights to stay at the hotel. However, neither `search_flights` or `search_hotels` can be called yet since the departure date of the flight and the check-in and check-out date of the hotel have not yet been provided.

## 3.3 Tool Calling

Once the agent determines which tools to invoke and receives the necessary parameters for each, it proceeds to make the appropriate calls. As shown in Figure 1, when the user provides the departure date for a flight to San Jose, the agent first calls the `search_flights` tool using the departure city, arrival city, and the specified departure date. Before it can proceed with booking accommodations, the agent must compute the check-out date for the hotel stay. To accomplish this, it calls the `adjust_date` tool. With both the check-in and check-out dates determined, along with the hotel location, the agent then calls the `search_hotels` tool. Finally, the results from both the flight and hotel searches are stored in the cache for potential future use.

## 3.4 Data Caching

We introduce a data caching mechanism that enables T1-AGENT to reuse the outputs of earlier tool calls when handling subsequent user requests. This reduces redundant computation and improves response efficiency in multi-turn interactions. After each user turn, any search results retrieved by a tool are cached for potential reuse in later turns. The `save_to_cache` tool is used to store these results. As illustrated in Figure 1, after the user provides the departure date for a flight to San Jose, the agent calls both `search_flights` and `search_hotels`, and then stores the results using the `save_to_cache` tool. Later, when the user requests flights from a specific airline, the agent retrieves the cached flight results using the `get_results_from_cache` tool and filters them using the `filter_flights` tool based on the user's airline preference. This caching approach helps avoid unnecessary API calls by reusing existing search results and applying filtering when appropriate.

**Example: Refining a Flight Search.**  Suppose a previous user query fetched flights from NYC to Boston, and the result was cached with the key `"flights_nyc_bos"`. Later, the user asks to see only flights priced under $500. The LLM, using the cache summary, generates the following plan:

```
flights = get_results_from_cache(key="flights_nyc_bos")
affordable_flights = filter_flights(prior_result=flights, budget=500)
```

This illustrates how the LLM composes new logic by combining previously retrieved results with downstream tools, without repeating expensive API operations. Our approach enables more efficient, coherent, and stateful plan generation in realistic, multi-turn assistant conversations.

# 4 Experimental Setup

## 4.1 Datasets

Each category contains 60 dialogue templates and a pool of 54 cities used to populate placeholder values. We create 15 training templates, 5 validation templates, and 40 test templates per category. Similarly, 13 cities are allocated for training, 4 for validation, and 37 for testing. Dialogues are randomly sampled within each partition. After partitioning, a script fills all placeholders in each template using the corresponding cities from the assigned split, ensuring no data leakage or overlap of templates and entities between the training and test sets. Each template is instantiated into 25

unique dialogues after substituting the placeholders in the dialogue templates with entities defined in the ontology, yielding a total of 1,500 fully generated dialogues.

For our experiments, we prepare two different test dataset splits: (1) `small`; (2) `large`. The `small` split is built for efficient and cost-effective evaluation and consists of 40 templates per domain and a total of 40 dialogues per domain. The `large` split is built using all 40 templates with a total of 1,000 dialogues for each domain.

## 4.2 Domain Adaptation Using SFT

We perform a simple instruction tuning with the train dataset in order to showcase that the performance on such complex conversational tool calling can be improved over zero or few-shot prompting. For this experiment, we train a `Llama 3.1 8B Instruct` model for one epoch on the training dataset. The training dataset is structured as a list of (`prompt, completions`) pairs. The SFT thereafter follows standard next token prediction with cross-entropy loss. We use LoRA [2] on 8 A100 SXM GPUs instead of full finetuning. For reproducibility, we use the widely adopted Huggingface TRL library [17] and include the exact command to replicate the training in Section E.

## 4.3 Inference Procedure

During inference, for each user turn in a dialogue, the T1-AGENT generates executable Python code that fulfills the user request at that point in the conversation. Before generating any code, the agent will check an **execution cache** to determine if a similar request has been previously resolved. If a cached result is available, the agent will write code that fetches and reuses the cached object to prevent redundant computation and tool invocation.

To do this, the agent constructs a prompt that includes the conversation history along with the current user turn. Instead of including the full execution cache, which can be large and token-intensive, the prompt incorporates a **summary** of the cached results. These summaries are generated using a deterministic, rule-based function that transforms each cached result into a concise representation. This approach significantly reduces the token load in the prompt, making it feasible to include relevant past results without exceeding model input limits.

This design enables the agent to consider past outcomes and reuse relevant information during code generation. Even in cases where the user's current query differs slightly from earlier queries, the agent is able to fetch a prior result from the cache and use that as a starting point for the current user turn. By introducing summarization at the *planning stage*, we shift caching responsibilities from the tool-execution layer to the agent's decision-making process. Our approach allows the agent to selectively reuse and adapt cached outputs, leading to improved latency and broader generalization.

The generated code is executed in a sandboxed environment, and the cache is updated with the new results on each user turn. Our dataset also includes the corresponding ground truth code and post-execution cache, which are used for performance evaluation.

## 5 Results and Analysis

In this section, we present the performance of various LLMs on both single-domain and multi-domain tasks. Single-domain tasks involve conversations focused exclusively on one domain, such as flights, hotels, restaurants, or attractions. In contrast, multi-domain tasks involve interactions spanning multiple domains, such as a user requesting both flight bookings and nearby hotel recommendations within the same dialogue.

## 5.1 Overall Results

We conduct evaluation on both the `small` and `large` dataset splits. For the `large` dataset, we compute the metrics averaged among all the domains for these models, all of which are open weight: `Llama 3.1 8B Instruct`, `Llama 3.1 8B Instruct SFT`, `Llama 3.3 70B Instruct`, `s1.1 32B`, and `Phi-4-reasoning-plus`. For the `small` dataset, we ran experiments for all of the models that are leveraged for the `large` dataset results with some additional open weight and proprietary models to provide additional insight.

As shown in Table 3, on the `large` dataset, the `Llama 3.1 8B Instruct` SFT model outperforms all other models across most evaluation metrics. However, it is surpassed in recall within both `Tool Calling` and `Parameter Matching` by the `Llama 3.3 70B Instruct`, and in `Code Execution Rate` accuracy by both the `Phi-4-reasoning-plus` and `Llama 3.3 70B Instruct` models.

In Table 4, for the `small` dataset, we evaluate the same set of models along with several additional open-weight and proprietary models to enable a more comprehensive comparison. While the `Llama 3.1 8B SFT` model achieves the highest overall performance across all metrics, it is outperformed in `Tool Calling` and `Parameter Matching` by all proprietary models, except GPT `4.1-nano`, within the open-weight model category. For `Code Execution Rate`, the `Llama 3.3 70B Instruct` model achieves accuracy comparable to proprietary models, although it still trails behind `Gemini 2.5 Pro`, `OpenAI o3`, and `GPT 5`. In the `Cache Summary` task, all proprietary models except GPT `4.1-nano` outperform the open-weight models. Interestingly, for the `Information Seeking` metrics, there is minimal performance difference between proprietary and open-weight models. In fact, the `Llama 3.1 8B Instruct SFT` achieves the highest scores on both SacreBLEU and BERTScore across all evaluated models.

| Domain | Tool Call | | | | Parameter Matching | | | | Code Exec. Rate | Information Seeking | | Cache Summary |
|---|---|---|---|---|---|---|---|---|---|---|---|---|
| | Acc. | Prec. | Rec. | F1 | Acc. | Prec. | Rec. | F1 | Acc. | SacreBLEU | BERTScore | EM |
| Phi-4-reasoning-plus | 53.65 | 75.20 | 63.68 | 68.58 | 35.51 | 60.45 | 48.04 | 51.31 | 86.37 | 27.49 | 81.03 | 40.59 |
| s1.1 32B | 72.06 | 79.66 | 88.08 | 83.38 | 55.56 | 63.72 | 80.42 | 70.95 | 60.76 | 17.85 | 79.90 | 54.29 |
| Llama 3.1 8B Instruct | 35.68 | 44.30 | 64.39 | 52.11 | 19.14 | 24.21 | 49.32 | 31.79 | 41.83 | 27.37 | 80.96 | 29.56 |
| Llama 3.1 8B Instruct SFT | **77.58** | **86.85** | 87.65 | **87.17** | **61.21** | **73.23** | 79.10 | **75.76** | 84.29 | **41.58** | **85.61** | **63.95** |
| Llama 3.3 70B Instruct | 67.39 | 76.63 | **88.17** | 79.72 | 53.75 | 61.83 | **83.17** | 67.74 | **91.43** | 30.64 | 82.66 | 57.83 |

Table 3: Overall results for LLMs using `large` dataset with few-shot in-context learning ($k = 13$).

| Domain | Tool Call | | | | Parameter Matching | | | | Code Exec. Rate | Information Seeking | | Cache Summary |
|---|---|---|---|---|---|---|---|---|---|---|---|---|
| | Acc. | Prec. | Rec. | F1 | Acc. | Prec. | Rec. | F1 | Acc. | SacreBLEU | BERTScore | EM |
| Open Weight | | | | | | | | | | | | |
| Phi-4-reasoning-plus | 54.27 | 75.45 | 64.45 | 69.14 | 35.98 | 60.20 | 48.84 | 51.84 | 86.45 | 27.26 | 80.91 | 40.91 |
| s1.1 1.5B | 5.93 | 5.93 | 11.11 | 7.73 | 0.00 | 0.00 | 0.00 | 0.00 | 3.95 | 0.00 | 0.00 | 26.98 |
| s1.1 3B | 28.05 | 37.36 | 48.05 | 41.28 | 13.99 | 16.95 | 39.34 | 23.16 | 5.98 | 0.00 | 0.00 | 26.78 |
| s1.1 7B | 30.45 | 46.03 | 52.66 | 40.09 | 19.39 | 32.48 | 37.98 | 27.78 | 5.37 | 12.74 | 69.32 | 27.18 |
| s1.1 32B | 71.15 | 79.04 | 87.41 | 82.72 | 54.60 | 62.83 | **79.78** | 70.01 | 60.65 | 17.65 | 79.72 | 52.75 |
| Llama 3.1 8B Instruct | 35.87 | 44.63 | 64.33 | 52.25 | 19.45 | 24.71 | 49.91 | 32.25 | 42.35 | 27.12 | 81.41 | 29.45 |
| Llama 3.1 8B Instruct SFT | 77.48 | 86.87 | 87.41 | 87.07 | 60.68 | 73.16 | 78.35 | 75.39 | 83.77 | **42.39** | **85.77** | **63.79** |
| Llama 3.1 70B Instruct | 72.28 | 80.06 | 87.49 | 83.49 | 59.16 | 68.79 | 80.61 | 73.59 | 77.62 | 32.46 | 83.20 | 53.94 |
| Llama 3.3 70B Instruct | 67.02 | 76.20 | 83.03 | 79.14 | 53.65 | 62.11 | 76.74 | 67.54 | **91.44** | 31.38 | 82.87 | 57.17 |
| Proprietary | | | | | | | | | | | | |
| Gemini 2.5 Pro | **89.32** | 93.49 | 95.18 | **94.28** | 73.68 | 82.49 | 87.04 | 84.63 | **94.46** | 7.11 | 74.44 | 76.11 |
| OpenAI o3 | 85.41 | 89.60 | 94.76 | 91.91 | 75.18 | 83.77 | 87.88 | 85.64 | 93.51 | 22.52 | 82.26 | 73.53 |
| OpenAI o4-mini | 86.47 | 91.88 | 93.44 | 92.59 | **77.31** | **86.91** | 87.32 | **87.05** | 89.00 | 22.93 | 81.52 | 72.43 |
| GPT 4.1 | 86.00 | 92.02 | 92.68 | 92.32 | 75.01 | 84.49 | 86.79 | 85.53 | 91.20 | **29.37** | **83.54** | 73.80 |
| GPT 4.1-mini | 83.99 | **93.77** | 88.64 | 91.09 | 72.96 | 83.92 | 84.70 | 84.20 | 87.29 | 20.83 | 81.12 | 67.27 |
| GPT 4.1-nano | 63.14 | 75.38 | 78.13 | 76.46 | 39.60 | 52.63 | 61.31 | 56.26 | 77.90 | 19.79 | 79.16 | 40.65 |
| GPT 5 | 87.52 | 90.92 | **95.76** | 93.14 | 76.54 | 84.63 | **88.71** | 86.51 | 94.45 | 28.15 | 81.94 | **76.25** |

Table 4: Overall results for LLMs using `small` dataset with few-shot in-context learning ($k = 13$).

## 5.2 Analysis

**The Impact of Domain Adaptation on Performance.** Figure 3 presents the Tool Call F1 and Parameter Matching F1 scores on the `large` dataset. We observe that SFT significantly improves domain adaptation for the smaller 8B model, enabling it to outperform its base version and even exceed the performance of the much larger 70B model by a considerable margin across all domains. This underscores the effectiveness of SFT in building robust and capable small models. Notably, while the 70B and non-fine-tuned 8B models were prompted to generate answers with explicit reasoning, the fine-tuned 8B variant, despite not being instructed to provide reasoning, achieved performance comparable to the 70B Instruct model. This highlights the strong potential of task-specific fine-tuning in enhancing model capabilities, even without explicit prompting.

**Few-Shot Evaluation.** As shown in Figure 4, we evaluate model performance on the *flights* domain under 0-shot, 5-shot, and 13-shot settings. Performance is notably poor in the 0-shot setting, with improvements observed in 5-shot and 13-shot configurations—though the gains plateau beyond 5 shots. Qualitative analysis suggests that without sufficient context, models continue to struggle with generalization in complex, multi-domain scenarios.

**Need for Complex Evaluation.** `Llama 3.3 70B Instruct` performs well on standard code generation tasks, we observe that it continues to struggle in complex, multi-turn scenarios involving

| Domain | Tool Call | | | | Parameter Matching | | | | Code Exec. Rate | Information Seeking | | Cache Summary |
|---|---|---|---|---|---|---|---|---|---|---|---|---|
| | Acc. | Prec. | Rec. | F1 | Acc. | Prec. | Rec. | F1 | Acc. | SacreBLEU | BERTScore | EM |
| Single-Domain | | | | | | | | | | | | |
| Flights | 25.12 | 30.90 | 57.34 | 40.16 | 15.57 | 17.34 | 60.30 | 26.94 | 26.24 | 14.41 | 82.36 | 36.63 |
| Hotels | 42.86 | 49.89 | 75.25 | 60.00 | 17.57 | 23.59 | 40.80 | 29.89 | 48.13 | **47.01** | 80.38 | 33.70 |
| Restaurants | 40.40 | 52.97 | 63.00 | 57.55 | **31.47** | **36.07** | **71.12** | **47.87** | **56.99** | 28.91 | 75.68 | 35.49 |
| Attractions | 33.03 | 48.49 | 50.90 | 49.66 | 14.11 | 25.11 | 24.36 | 24.73 | 40.40 | N/A | N/A | **43.87** |
| Multi-Domain. F: Flights, H: Hotels, R: Restaurants, A: Attractions | | | | | | | | | | | | |
| F-H | 33.09 | 40.89 | 63.44 | 49.73 | 16.11 | 19.28 | 49.48 | 27.75 | 38.43 | 26.54 | 82.00 | 21.85 |
| H-R | **51.06** | **59.21** | **78.77** | **67.61** | 26.62 | 33.52 | 56.37 | 42.05 | 51.82 | 28.60 | **84.07** | 9.95 |
| H-A | 38.75 | 45.46 | 72.39 | 55.85 | 16.11 | 20.02 | 45.12 | 27.74 | 48.88 | 26.03 | 81.77 | 32.75 |
| F-H-A | 27.76 | 36.24 | 54.24 | 43.45 | 16.21 | 21.18 | 40.88 | 27.90 | 28.76 | 14.15 | 78.58 | 24.37 |
| F-H-R | 29.04 | 34.65 | 64.19 | 45.00 | 18.53 | 21.77 | 55.44 | 31.26 | 36.79 | 33.28 | 82.84 | 27.47 |

Table 5: Overall results using `Llama 3.1 8B Instruct` with few-shot in-context learning ($k = 13$) on `large` dataset.

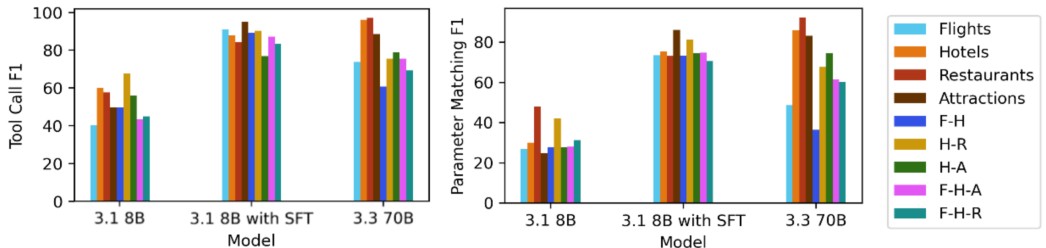

Figure 3: **Left:** Tool Call F1 and **Right:** Parameter Matching F1 on `large` dataset.

advanced planning. The T1 dataset is designed to fill this gap by serving as a benchmark for evaluating model performance in such challenging settings.

**Open-weight vs. Proprietary Models.** With the `small` dataset, we experiment with both `open-weight` and `proprietary` models. From Figure 5, among open-weight models, we observe that the `Llama 3.1 8B Instruct SFT` model performs the best based on the tool calling F1 score however it is still out-performed by almost all of the proprietary models except for `GPT 4.1-nano`. This indicates that while fine-tuning an open-weight model does improve the performance significantly, there is a limit in the amount of improvement possible that cannot match the results from the top proprietary models.

# 6 Related Work

## 6.1 Large Language Model Agents

LLM-based agents have emerged as foundational components in AI systems, capable of performing complex, multi-step tasks through reasoning, memory integration, and tool use. These agents often combine a pre-trained LLM with structured modules such as long-term memory, tool calling capabilities, and self-reflective feedback loops. Frameworks such as AutoGPT [21], and AgentGPT allow LLMs to autonomously decompose user goals into subgoals and execute them sequentially using external APIs. More structured systems like HuggingGPT [13], CrewAI, and AutoGen [19] facilitate collaboration between multiple LLM agents, each specializing in roles such as planning, execution, or critique.

Despite significant progress, planning within task-oriented dialogue systems—particularly over long horizons—remains a fundamental challenge. Previous paradigms such as *plan-observe-execute* (e.g., ReAct [22], ADaPT [10]) enable the model to interleave tool calls with reasoning steps. However, most frameworks focus on linear execution paths where each step invokes a single tool. These approaches often lack the ability to manage inter-tool dependencies, reuse intermediate results, or revise plans based on partial tool failures. Further, a lot of these systems performs single turn planning, i.e., the user of the system submits a request with all the information in the first turn. Multi-turn conversational planning is a nascent field. For example, in multi-turn workflows like conversational

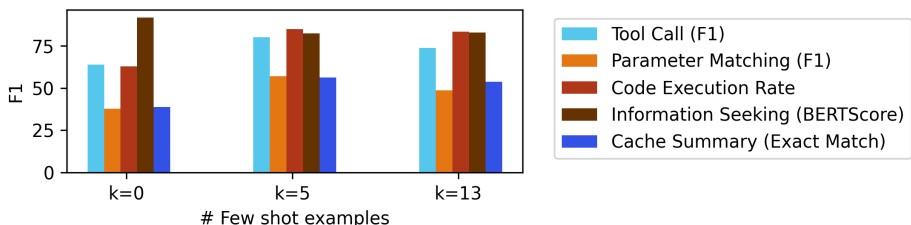

Figure 4: Few-shot performance on `flight` using `Llama 3.3 70B Instruct` on `large` dataset.

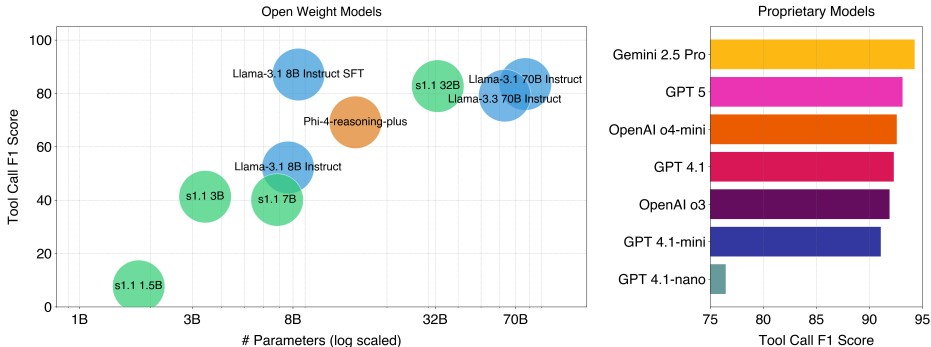

Figure 5: Tool Calling F1 score between open weight and proprietary models on `small` dataset.

trip planning, agents need to collect information across multiple steps, then plan and coordinate across tools like flight search, visa information, and calendar APIs—something most current systems struggle to handle.

## 6.2 Tool-Based Agents and Benchmarks

Tool usage extends the scope of LLM capabilities beyond language modeling [16, 12] to real-world actionability, enabling interactions with APIs, web tools, and external software. To evaluate this ability, several benchmarks have been proposed, including APIBank [4], Tau-Bench [23], GAIA [6], ALFWorld [14], GTA [18], TravelPlanner [20], and ToolBench [11]. Although these benchmarks each emphasize different strengths, such as breadth of API coverage, realism, or reasoning complexity, they often treat tool use as a series of isolated atomic actions without requiring coordinated planning between multiple tools.

In contrast, our work introduces T1, a tool-driven benchmark designed to evaluate LLM agents in multi-turn, multi-domain dialogue settings with inter-tool dependencies. T1 features an integrated caching mechanism that supports both short- and long-term memory of tool call results, allowing agents to make intelligent decisions about whether to replan or reuse cached outputs. Unlike prior benchmarks, it tests an agent's ability to perform dynamic replanning, handle branching workflows, and compose tools in a realistic, dialog-driven environment. Thus, T1 not only challenges existing tool-use agents but also provides a diagnostic sandbox for evaluating the reasoning capabilities of open-weight LLMs under realistic constraints.

## 7 Conclusion

We introduce T1, a dataset for evaluating planning, reasoning, and tool use in LLM-based agents through complex, multi-turn dialogues. By incorporating inter-tool dependencies, dynamic replanning, and caching, it supports rigorous evaluation across single- and multi-domain tasks. Experiments with T1-AGENT show that fine-tuned models often outperform much larger untuned ones, underscoring the value of task-specific adaptation. We also find that stronger reasoning leads to more effective tool use. We hope T1 will serve as a valuable resource for developing more capable, tool-augmented language agents.

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

## Acknowledgments

We would like to thank Xiuzhu Lin and Ashley Trick for the valuable discussion on the paper.


# A    Limitations

In this work, we focus on constructing and introducing a new dataset as a benchmark for evaluating agentic workflows in multi-turn conversational dialogue settings, with an emphasis on tool calling for planning. Our evaluations are limited to open-weight models, as proprietary models are not included due to resource constraints. We use `Llama 3.1 8B Instruct`, `Llama 3.3 70B Instruct`, `s1.1 32B`, and `Phi-4-reasoning-plus`. We will release T1 dataset publicly and hope it will encourage future research that includes evaluations on proprietary models as well.

# B    Evaluation Protocol

We evaluate the model's performance at every user turn by comparing the model's output with the ground truth across the following facets.

**Tool Call.**    We evaluate the correctness of each tool call from the generated code against the ground truth code by comparing the number of times each tool is called using four metrics: accuracy, precision, recall, and F1.

**Parameter Matching.**    For each tool call in the ground truth, we identify the corresponding tool call in the generated code with the same name and the highest parameter overlap. We then calculate the accuracy, precision, recall, and F1 for the identified match. Parameter values are standardized to enable robust comparison; for example, lists are treated as sets to make them insensitive to ordering. Certain tool calls such as `save_to_cache`, `get_results_from_cache`, and `seek_information` are excluded from this evaluation, as they depend on external artifacts (e.g., intermediate dataframes) and variable keys or identifiers that cannot be reliably matched. Similarly, certain parameter names (e.g., `prior_results`) are also excluded.

**Code Execution Success Rate.**    We calculate the percentage of instances in which the model generates a code when expected and the code is executable without any errors in a sandbox environment.

**Handling Information Seeking.**    If both the model output and the ground truth include `seek_information`, we evaluate the similarity of the strings inside the function using Sacre-BLEU [9] and BERTScore F1 [24], capturing both sub-word overlap and semantic similarity of the requested information.

**Cache Summary.**    To assess how well the model serves complete requests, we compare the execution cache summary results of the generated code and ground truth using exact match (EM) to determine if the model-generated solution is functionally equivalent to the ground truth.

# C  Additional Results

## C.1  Results on the cache usage across domains and models

Table 6 shows the average success rate of the cache hit. We define a cache hit is when the plan requires a cache read, and the cache indeed has the required results. We evaluate the same set of models along with several additional open-weight and proprietary models to enable a more comprehensive comparison on the `small` dataset. Here clearly the domain adapted Llama 3.1 8B Instruct SFT models has the highest average overall. The large proprietary models are also highly performant, however this suggests the even small domain adapted models can beat large general purpose LLMs.

In Table 7, we further show some statistics of the cache usage. We have the ground truth statistics as `Reference`. We show the average tool call per turn of conversation, average times we save in the cache and average number of times we retrieve from the cache every turn across all the domains. We observe that the performance of the SFT model is the closest to the ground truth with the `OpenAI o4-mini` coming in close.

| Model | Flights | Hotels | Restaurants | Attractions | F-H | H-R | H-A | F-H-A | F-H-R | Avg. |
|---|---|---|---|---|---|---|---|---|---|---|
| Multi-Domain. F: Flights, H: Hotels, R: Restaurants, A: Attractions | | | | | | | | | | |
| Open Weight | | | | | | | | | | |
| Phi-4-reasoning-plus | 17.93 | **98.90** | **97.43** | 31.02 | 54.93 | 61.86 | 88.17 | 29.13 | 47.89 | 58.58 |
| s1.1 32B | 54.75 | 53.53 | 49.70 | 49.47 | 59.52 | 56.73 | 47.79 | 63.04 | 64.14 | 55.41 |
| Llama 3.1 8B Instruct | 1.36 | 30.12 | 7.12 | 0.28 | 17.30 | 55.25 | 52.00 | 17.03 | 22.71 | 22.57 |
| Llama 3.1 8B Instruct SFT | 82.04 | **98.90** | 96.63 | **98.53** | **88.23** | 75.58 | 81.39 | 92.28 | 86.07 | **88.85** |
| Llama 3.3 70B Instruct | 31.87 | 94.70 | 96.54 | 28.32 | 40.03 | 78.84 | 90.54 | 46.22 | 46.54 | 61.51 |
| Proprietary | | | | | | | | | | |
| Gemini 2.5 Pro | **86.29** | 97.23 | 97.23 | 41.60 | 87.27 | 88.68 | **96.79** | **99.84** | 94.27 | 87.58 |
| OpenAI o4-mini | 81.13 | 92.11 | **97.43** | 55.45 | 81.53 | **94.47** | 96.61 | 92.14 | **94.92** | 87.31 |

Table 6: Average Success Rate on `get_results_from_cache`.

| Model | Success Rate (Retrieve Cache) | Avg. Tool Call | Avg. Save Cache | Avg. Retrieve Cache |
|---|---|---|---|---|
| Reference | 100 | 2.03 | 0.70 | 0.43 |
| Open Weight | | | | |
| Phi-4-reasoning-plus | 58.58 | 1.57 | 0.49 | 0.36 |
| s1.1 32B | 55.41 | 1.66 | 0.59 | 0.35 |
| Llama 3.1 8B Instruct | 22.57 | 1.876 | 0.61 | 0.24 |
| Llama 3.1 8B Instruct SFT | **88.85** | 2.11 | 0.77 | 0.47 |
| Llama 3.3 70B Instruct | 61.51 | 2.18 | 0.80 | 0.38 |
| Proprietary | | | | |
| Gemini 2.5 Pro | 87.58 | 2.05 | 0.73 | 0.40 |
| OpenAI o4-mini | 87.31 | **2.03** | **0.70** | **0.44** |

Table 7: Statistics of cache usage.

## C.2 Detailed Results on Large Dataset

In this section, we show the detailed results on the `large` dataset. Table 8 shows the results for `Llama 3.1 8B Instruct` with Domain Adaptation and Table 9 shows the results for `Llama 3.3 70B Instruct`.

Additionally, we also experiment with medium-sized reasoning models to see how they perform. We work with `s1.1 32B` [7] and `Phi-4-reasoning-plus 14B` [1]. As shown in Table 10 and Table 11, the `s1.1 32B` model mostly outperforms the `Phi-4-reasoning-plus` model when it comes to Tool Call and Parameter Matching metrics.

| Domain | Tool Call | | | | Parameter Matching | | | | Code Exec. Rate | Information Seeking | | Cache Summary |
|---|---|---|---|---|---|---|---|---|---|---|---|---|
| | Acc. | Prec. | Rec. | F1 | Acc. | Prec. | Rec. | F1 | Acc. | SacreBLEU | BERTScore | EM |
| Single-Domain | | | | | | | | | | | | |
| Flights | 83.51 | 89.98 | 92.08 | 91.02 | 58.02 | 66.38 | 82.18 | 73.44 | 85.26 | 39.19 | 86.44 | 62.07 |
| Hotels | 78.18 | 84.87 | 90.84 | 87.75 | 60.40 | 67.29 | **85.53** | 75.32 | 72.13 | 48.30 | 80.91 | 71.35 |
| Restaurants | 72.76 | 80.40 | 88.44 | 84.23 | 57.49 | 66.69 | 80.64 | 73.00 | 95.16 | **95.14** | **98.63** | 65.32 |
| Attractions | **90.50** | **97.87** | 92.33 | **95.02** | **75.54** | **91.34** | 81.37 | **86.07** | **98.15** | N/A | N/A | **83.00** |
| Multi-Domain. F: Flights, H: Hotels, R: Restaurants, A: Attractions | | | | | | | | | | | | |
| F-H | 80.27 | 91.31 | 86.91 | 89.05 | 57.76 | 74.51 | 71.99 | 73.23 | 89.25 | 24.04 | 80.63 | 57.01 |
| H-R | 82.02 | 91.03 | 89.23 | 90.12 | 68.44 | 77.95 | 84.86 | 81.26 | 83.67 | 29.12 | 87.58 | 64.42 |
| H-A | 62.30 | 72.67 | 81.35 | 76.77 | 59.22 | 68.73 | 81.05 | 74.39 | 71.41 | 27.87 | 82.80 | 65.70 |
| F-H-A | 77.19 | 87.50 | 86.76 | 87.13 | 59.63 | 74.16 | 75.27 | 74.71 | 81.15 | 22.65 | 83.45 | 52.76 |
| F-H-R | 71.53 | 86.05 | 80.91 | 83.40 | 54.39 | 71.99 | 68.99 | 70.45 | 82.47 | 46.32 | 84.40 | 53.88 |

Table 8: Overall results using `Llama 3.1 8B Instruct` SFT. on `large` dataset.

| Domain | Tool Call | | | | Parameter Matching | | | | Code Exec. Rate | Information Seeking | | Cache Summary |
|---|---|---|---|---|---|---|---|---|---|---|---|---|
| | Acc. | Prec. | Rec. | F1 | Acc. | Prec. | Rec. | F1 | Acc. | SacreBLEU | BERTScore | EM |
| Single-Domain | | | | | | | | | | | | |
| Flights | 58.42 | 78.26 | 69.75 | 73.76 | 32.11 | 40.51 | 60.76 | 48.61 | 83.32 | 20.25 | 82.90 | 53.80 |
| Hotels | 92.42 | 96.03 | 96.10 | 96.07 | 75.33 | 80.85 | **91.69** | 85.93 | 98.55 | **46.98** | 80.27 | 75.00 |
| Restaurants | **94.23** | **96.63** | **97.43** | **97.03** | **85.51** | **93.62** | 90.80 | **92.19** | 99.54 | 37.90 | 79.95 | **90.78** |
| Attractions | 79.14 | 92.16 | 84.86 | 88.36 | 71.17 | 87.04 | 79.60 | 83.16 | 63.70 | N/A | N/A | 78.59 |
| Multi-Domain. F: Flights, H: Hotels, R: Restaurants, A: Attractions | | | | | | | | | | | | |
| F-H | 43.48 | 53.73 | 69.52 | 60.61 | 22.28 | 25.94 | 61.24 | 36.44 | 95.02 | 27.39 | 83.42 | 37.15 |
| H-R | 60.50 | 70.91 | 80.46 | 78.29 | 50.94 | 60.08 | 76.99 | 67.50 | **99.60** | 22.97 | **84.11** | 38.43 |
| H-A | 64.91 | 68.93 | 91.75 | 78.72 | 59.24 | 66.59 | 84.29 | 74.40 | 97.01 | 30.02 | 82.71 | 57.03 |
| F-H-A | 60.46 | 70.77 | 80.59 | 75.36 | 44.27 | 51.19 | 76.58 | 61.37 | 94.81 | 19.16 | 82.27 | 44.29 |
| F-H-R | 52.97 | 62.22 | 78.07 | 69.25 | 42.93 | 50.66 | 73.76 | 60.07 | 91.30 | 40.45 | 85.62 | 45.37 |

Table 9: Overall results using `Llama 3.3 70B Instruct` with few-shot in-context learning ($k = 13$) on `large` dataset.

| Domain | Tool Call | | | | Parameter Matching | | | | Code Exec. Rate | Information Seeking | | Cache Summary |
|---|---|---|---|---|---|---|---|---|---|---|---|---|
| | Acc. | Prec. | Rec. | F1 | Acc. | Prec. | Rec. | F1 | Acc. | SacreBLEU | BERTScore | EM |
| Single-Domain | | | | | | | | | | | | |
| Flights | 77.59 | 87.87 | 86.90 | 87.38 | 49.53 | 58.04 | 77.15 | 66.25 | 57.20 | **29.65** | 83.12 | 57.46 |
| Hotels | 89.33 | 95.33 | 93.41 | **94.36** | 65.27 | 70.52 | 89.77 | 78.99 | 63.95 | 18.07 | 79.56 | 57.23 |
| Restaurants | 87.77 | 92.65 | **94.35** | 93.49 | **71.59** | **77.69** | 90.11 | **83.44** | 69.82 | 7.16 | 69.91 | **69.22** |
| Attractions | 65.76 | 80.16 | 78.54 | 79.34 | 44.92 | 53.04 | 74.59 | 62.00 | 72.57 | N/A | N/A | 63.87 |
| Multi-Domain. F: Flights, H: Hotels, R: Restaurants, A: Attractions | | | | | | | | | | | | |
| F-H | 69.87 | 78.42 | 86.51 | 82.27 | 42.38 | 50.39 | 72.71 | 59.53 | 66.89 | 9.72 | 78.43 | 47.49 |
| H-R | 59.10 | 65.62 | 85.60 | 74.29 | 51.91 | 58.54 | 82.10 | 68.35 | 49.72 | 19.69 | **85.40** | 32.55 |
| H-A | 63.46 | 67.81 | 90.83 | 77.65 | 65.22 | 72.53 | 86.60 | 78.95 | 44.39 | 21.41 | 81.95 | 56.64 |
| F-H-A | 68.41 | 75.27 | 88.25 | 81.24 | 50.21 | 61.04 | 73.88 | 66.85 | 60.86 | 27.00 | 82.48 | 50.42 |
| F-H-R | 67.22 | 73.80 | 88.30 | 80.40 | 58.97 | 71.66 | 76.91 | 74.19 | 61.47 | 10.08 | 78.32 | 53.74 |

Table 10: Overall results using `s1.1 32B` with few-shot in-context learning ($k = 13$) on `large` dataset.

| Domain | Tool Call | | | | Parameter Matching | | | | Code Exec. Rate | Information Seeking | | Cache Summary |
| | Acc. | Prec. | Rec. | F1 | Acc. | Prec. | Rec. | F1 | Acc. | SacreBLEU | BERTScore | EM |
| --- | --- | --- | --- | --- | --- | --- | --- | --- | --- | --- | --- | --- |
| Single-Domain | | | | | | | | | | | | |
| Flights | 36.66 | 67.30 | 44.61 | 53.66 | 15.74 | 56.10 | 17.95 | 27.20 | 72.76 | 13.91 | 81.60 | 39.73 |
| Hotels | **83.96** | **91.74** | **90.82** | **91.28** | **53.22** | 63.09 | **77.30** | **69.47** | 93.05 | **46.95** | 80.47 | 46.52 |
| Restaurants | 64.13 | 83.60 | 73.36 | 78.14 | 47.10 | 61.89 | 66.34 | 64.04 | 93.49 | 28.27 | 69.74 | 50.53 |
| Attractions | 35.78 | 61.67 | 46.02 | 52.71 | 22.90 | 66.92 | 25.82 | 37.27 | 66.25 | N/A | N/A | 50.79 |
| Multi-Domain. F: Flights, H: Hotels, R: Restaurants, A: Attractions | | | | | | | | | | | | |
| F-H | 57.79 | 88.12 | 62.67 | 73.25 | 29.91 | 59.20 | 37.68 | 46.05 | 83.37 | 27.96 | 83.02 | 32.53 |
| H-R | 54.07 | 70.71 | 69.67 | 70.18 | 35.42 | 50.91 | 53.79 | 52.31 | **96.04** | 23.19 | **84.83** | 21.72 |
| H-A | 65.81 | 78.69 | 80.08 | 79.38 | 48.58 | 61.78 | 69.45 | 65.39 | **96.04** | 27.92 | 82.97 | **53.52** |
| F-H-A | 37.83 | 62.18 | 49.14 | 54.90 | 32.73 | **64.59** | 39.89 | 49.32 | 84.49 | 15.42 | 81.00 | 32.33 |
| F-H-R | 46.79 | 72.75 | 56.74 | 63.75 | 33.99 | 59.56 | 44.18 | 50.73 | 91.86 | 36.30 | 84.63 | 37.65 |

Table 11: Overall results using `Phi-4-reasoning-plus` with few-shot in-context learning ($k = 13$) on `large` dataset.

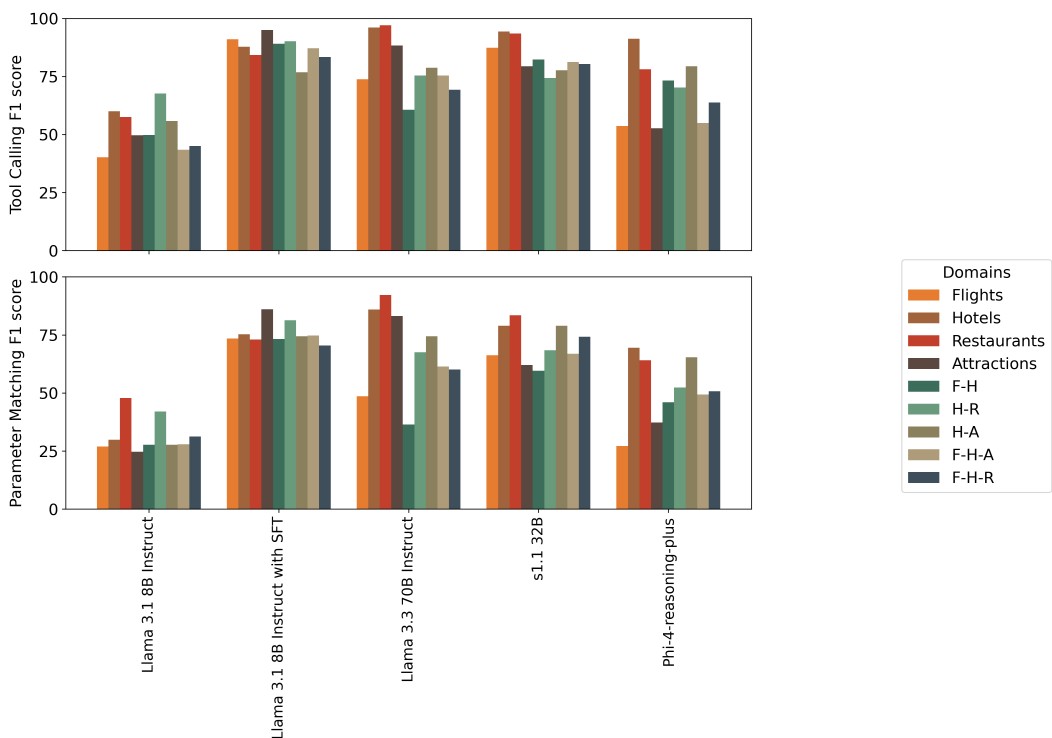

Figure 6: **Above:** Tool Call F1 performance and **Below:** Parameter Matching F1 performance on `large` dataset.

### C.3 Detailed Results on Small Dataset

In this section, we present detailed results on the `small` dataset. Tables 12 through 27 report the performance of various models, including `Gemini 2.5 Pro` (Table 12), `GPT-5` (Table 13), `GPT-4.1-nano` (Table 14), `OpenAI o3` (Table 15), `GPT-4.1-mini` (Table 16), `OpenAI o4-mini` (Table 17), and `GPT-4.1` (Table 18). `s1.1 32B` (Table 19), `s1.1 7B` (Table 20), `s1.1 3B` (Table 21), `s1.1 1.5B` (Table 22), `Phi-4-reasoning-plus` (Table 23), `Llama 3.1 8B Instruct` (Table 24), `Llama 3.1 8B Instruct SFT` (Table 25), `Llama 3.1 70B Instruct` (Table 26), and `Llama 3.3 70B Instruct` (Table 27).

Figures 7 through 9 illustrate the comparative performance of open-weight and proprietary models under few-shot in-context learning ($k = 13$). Specifically, Figure 7 reports the Parameter Matching F1 score, Figure 8 presents Code Execution Rate Accuracy, and Figure 9 shows the Cache Summary Exact Match score.

| Domain | Tool Call | | | | Parameter Matching | | | | Code Exec. Rate | Information Seeking | | Cache Summary |
|---|---|---|---|---|---|---|---|---|---|---|---|---|
| | Acc. | Prec. | Rec. | F1 | Acc. | Prec. | Rec. | F1 | Acc. | SacreBLEU | BERTScore | EM |
| Single-Domain | | | | | | | | | | | | |
| Attractions | 85.05 | 91.67 | 92.18 | 91.92 | 68.15 | 79.31 | 82.88 | 81.06 | 73.95 | N/A | N/A | 81.51 |
| Flights | 91.24 | 94.74 | 96.12 | 95.42 | 69.52 | 78.96 | 85.31 | 82.02 | 82.93 | 6.29 | 76.07 | 76.22 |
| Hotels | **97.58** | **99.18** | 98.37 | **98.78** | 79.55 | 84.84 | 92.73 | 88.61 | 98.75 | 9.13 | 73.90 | 80.00 |
| Restaurants | 95.87 | 98.37 | 97.42 | 97.89 | 84.69 | **92.74** | 90.71 | 91.71 | **100.00** | 4.34 | 63.70 | **91.77** |
| Multi-Domain. F: Flights, H: Hotels, R: Restaurants, A: Attractions | | | | | | | | | | | | |
| FH | 85.90 | 91.67 | 93.17 | 92.41 | 65.92 | 73.91 | 85.92 | 79.46 | 98.29 | 6.57 | 75.84 | 67.95 |
| FHA | 92.47 | 94.35 | 97.89 | 96.09 | 73.07 | 81.41 | 87.71 | 84.44 | 98.97 | 7.43 | 76.16 | 71.28 |
| FHR | 86.12 | 95.15 | 90.07 | 92.54 | 68.25 | 83.48 | 78.90 | 81.13 | 97.67 | 6.81 | 75.76 | 73.84 |
| HA | 88.87 | 89.69 | **98.98** | 94.11 | **87.09** | 90.93 | **95.37** | **93.10** | 99.54 | **10.27** | **76.92** | 80.73 |
| HR | 80.82 | 86.56 | 92.41 | 89.39 | 66.87 | 76.81 | 83.78 | 80.15 | **100.00** | 6.02 | 77.21 | 61.64 |

Table 12: Overall results using `Gemini 2.5 Pro` with few-shot in-context learning ($k = 13$) on `small` dataset.

| Domain | Tool Call | | | | Parameter Matching | | | | Code Exec. Rate | Information Seeking | | Cache Summary |
|---|---|---|---|---|---|---|---|---|---|---|---|---|
| | Acc. | Prec. | Rec. | F1 | Acc. | Prec. | Rec. | F1 | Acc. | SacreBLEU | BERTScore | EM |
| Single-Domain | | | | | | | | | | | | |
| Attractions | 95.88 | 98.19 | 97.60 | 97.90 | 84.30 | 89.47 | 93.58 | 91.48 | 67.23 | N/A | N/A | 89.92 |
| Flights | 94.70 | 97.28 | 97.28 | 97.28 | 72.04 | 83.75 | 83.75 | 83.75 | 85.98 | **51.61** | **89.11** | 73.78 |
| Hotels | 97.59 | **99.18** | 98.38 | 98.78 | 79.85 | 82.87 | **95.64** | 88.80 | **100.00** | 41.85 | 80.43 | 76.88 |
| Restaurants | **97.78** | 98.40 | **99.35** | **98.88** | **86.73** | **92.90** | 92.90 | **92.90** | **100.00** | 12.59 | 75.28 | **91.77** |
| Multi-Domain. F: Flights, H: Hotels, R: Restaurants, A: Attractions | | | | | | | | | | | | |
| FH | 80.36 | 88.45 | 89.79 | 89.11 | 60.86 | 72.05 | 79.68 | 75.67 | 99.57 | 11.74 | 78.24 | 70.51 |
| FHA | 84.58 | 89.60 | 93.80 | 91.65 | 73.28 | 83.49 | 85.69 | 84.58 | 99.49 | 31.32 | 85.93 | 69.23 |
| FHR | 78.48 | 86.30 | 89.65 | 87.94 | 73.71 | 89.48 | 80.70 | 84.86 | 98.26 | 15.74 | 78.89 | 70.93 |
| HA | 80.40 | 80.93 | 99.19 | 89.13 | 82.29 | 86.56 | 94.34 | 90.28 | 99.54 | 28.84 | 82.82 | 73.39 |
| HR | 77.91 | 79.95 | 96.83 | 87.59 | 75.83 | 81.12 | 92.08 | 86.26 | **100.00** | 31.48 | 84.80 | 69.86 |

Table 13: Overall results using `GPT-5` with few-shot in-context learning ($k = 13$) on `small` dataset.

| Domain | Tool Call | | | | Parameter Matching | | | | Code Exec. Rate | Information Seeking | | Cache Summary |
|---|---|---|---|---|---|---|---|---|---|---|---|---|
| | Acc. | Prec. | Rec. | F1 | Acc. | Prec. | Rec. | F1 | Acc. | SacreBLEU | BERTScore | EM |
| Single-Domain | | | | | | | | | | | | |
| Attractions | 80.00 | 88.64 | 89.14 | 88.89 | 41.73 | 59.79 | 58.00 | 58.88 | 80.67 | N/A | N/A | 54.62 |
| Flights | 66.67 | 90.03 | 71.98 | 80.00 | 36.41 | 53.09 | 53.68 | 53.38 | 75.61 | 12.31 | 80.23 | 43.29 |
| Hotels | **89.63** | **94.13** | **94.93** | **94.53** | **53.16** | **61.27** | **80.06** | **69.41** | 82.50 | **30.30** | 80.73 | 43.75 |
| Restaurants | 73.81 | 84.35 | 85.52 | 84.93 | 51.13 | 58.62 | 80.00 | 67.66 | 79.75 | 10.54 | 66.15 | **61.39** |
| Multi-Domain. F: Flights, H: Hotels, R: Restaurants, A: Attractions | | | | | | | | | | | | |
| FH | 55.04 | 66.79 | 75.77 | 71.00 | 32.29 | 45.59 | 52.54 | 48.82 | 70.94 | 17.62 | 77.74 | 30.34 |
| FHA | 44.23 | 62.01 | 60.67 | 61.33 | 29.44 | 49.82 | 41.84 | 45.48 | 77.44 | 13.51 | 80.02 | 30.77 |
| FHR | 47.81 | 58.50 | 72.35 | 64.69 | 40.18 | 54.43 | 60.54 | 57.33 | 76.74 | 26.22 | 79.97 | 36.63 |
| HA | 58.63 | 69.05 | 79.54 | 73.92 | 42.10 | 52.49 | 68.01 | 59.25 | 80.73 | 22.96 | 82.53 | 48.17 |
| HR | 52.48 | 64.88 | 73.30 | 68.84 | 29.93 | 38.60 | 57.14 | 46.08 | 76.71 | 24.85 | **85.93** | 16.89 |

Table 14: Overall results using `GPT-4.1-nano` with few-shot in-context learning ($k = 13$) on `small` dataset.

| Domain | Tool Call | | | | Parameter Matching | | | | Code Exec. Rate | Information Seeking | | Cache Summary |
|---|---|---|---|---|---|---|---|---|---|---|---|---|
| | Acc. | Prec. | Rec. | F1 | Acc. | Prec. | Rec. | F1 | Acc. | SacreBLEU | BERTScore | EM |
| Single-Domain | | | | | | | | | | | | |
| Attractions | 86.67 | 92.86 | 92.86 | 92.86 | 69.12 | 77.05 | 87.04 | 81.74 | 77.31 | N/A | N/A | 83.19 |
| Flights | 93.36 | 96.57 | 96.57 | 96.57 | 70.90 | 82.08 | 83.88 | 82.97 | 82.32 | **50.83** | **89.74** | 73.78 |
| Hotels | **97.33** | **99.18** | 98.11 | **98.64** | 81.19 | 84.54 | 95.35 | 89.62 | **100.00** | 18.29 | 80.08 | 76.88 |
| Restaurants | 97.09 | 98.36 | **98.68** | 98.52 | **86.01** | **93.26** | 91.71 | **92.48** | 98.73 | 8.41 | 70.71 | **91.77** |
| Multi-Domain. F: Flights, H: Hotels, R: Restaurants, A: Attractions | | | | | | | | | | | | |
| FH | 81.20 | 89.16 | 90.09 | 89.62 | 62.20 | 73.60 | 80.06 | 76.69 | 95.73 | 9.54 | 79.67 | 68.80 |
| FHA | 85.71 | 88.48 | 96.48 | 92.31 | 78.61 | 86.81 | 89.27 | 88.02 | 96.92 | 36.93 | 87.83 | 67.69 |
| FHR | 74.59 | 86.10 | 84.80 | 85.44 | 70.52 | 88.39 | 77.72 | 82.71 | 96.51 | 11.45 | 79.77 | 69.19 |
| HA | 77.65 | 79.79 | 96.65 | 87.42 | 78.45 | 85.46 | 90.54 | 87.93 | 97.25 | 22.83 | 83.33 | 70.64 |
| HR | 75.08 | 75.90 | 98.58 | 85.77 | 79.60 | 82.78 | **95.39** | 88.64 | 96.80 | 21.84 | 86.98 | 59.82 |

Table 15: Overall results using `OpenAI o3` with few-shot in-context learning ($k = 13$) on `small` dataset.

| Domain | Tool Call | | | | Parameter Matching | | | | Code Exec. Rate | Information Seeking | | Cache Summary |
|---|---|---|---|---|---|---|---|---|---|---|---|---|
| | Acc. | Prec. | Rec. | F1 | Acc. | Prec. | Rec. | F1 | Acc. | SacreBLEU | BERTScore | EM |
| Single-Domain | | | | | | | | | | | | |
| Attractions | 86.90 | 94.81 | 91.25 | 92.99 | 71.77 | 80.91 | 86.41 | 83.57 | 71.43 | N/A | N/A | 77.31 |
| Flights | 87.11 | 93.58 | 92.64 | 93.11 | 70.36 | 78.26 | 87.45 | 82.60 | 72.56 | **32.75** | 84.82 | 60.98 |
| Hotels | **96.75** | **99.09** | **97.61** | **98.35** | 81.28 | 86.61 | **92.97** | 89.68 | 91.25 | 20.32 | 79.39 | 68.12 |
| Restaurants | 94.88 | 97.54 | 97.20 | 97.37 | **82.14** | 88.46 | 92.00 | **90.20** | 94.94 | 13.72 | 71.47 | 88.61 |
| Multi-Domain. F: Flights, H: Hotels, R: Restaurants, A: Attractions | | | | | | | | | | | | |
| FH | 76.56 | 89.39 | 84.22 | 86.73 | 62.16 | 76.11 | 77.22 | 76.66 | 81.20 | 12.35 | 80.54 | 53.85 |
| FHA | 81.28 | 94.94 | 84.96 | 89.67 | 71.74 | 86.20 | 81.05 | 83.55 | 87.69 | 30.40 | 84.29 | 61.54 |
| FHR | 75.62 | 92.19 | 80.80 | 86.12 | 68.43 | 87.45 | 75.88 | 81.26 | 94.77 | 12.87 | 80.33 | 64.53 |
| HA | 85.04 | 96.29 | 87.92 | 91.92 | 81.29 | **90.16** | 89.21 | 89.68 | 94.95 | 22.13 | 82.39 | **78.90** |
| HR | 71.75 | 86.08 | 81.17 | 83.55 | 67.50 | 81.08 | 80.12 | 80.59 | **96.80** | 22.08 | **85.77** | 51.60 |

Table 16: Overall results using `GPT-4.1-mini` with few-shot in-context learning ($k = 13$) on `small` dataset.

| Domain | Tool Call | | | | Parameter Matching | | | | Code Exec. Rate | Information Seeking | | Cache Summary |
|---|---|---|---|---|---|---|---|---|---|---|---|---|
| | Acc. | Prec. | Rec. | F1 | Acc. | Prec. | Rec. | F1 | Acc. | SacreBLEU | BERTScore | EM |
| Single-Domain | | | | | | | | | | | | |
| Attractions | 32.43 | 42.86 | 57.14 | 48.98 | 20.83 | 23.81 | 62.50 | 34.48 | 14.29 | N/A | N/A | **39.50** |
| Flights | 7.69 | 14.29 | 14.29 | 14.29 | 0.00 | 0.00 | 0.00 | 0.00 | 6.10 | N/A | N/A | 37.20 |
| Hotels | **73.08** | **82.61** | **86.36** | **84.44** | 25.00 | 35.29 | 46.15 | 40.00 | 3.12 | N/A | N/A | 26.25 |
| Restaurants | 26.23 | 39.02 | 44.44 | 41.56 | **32.05** | **35.71** | **75.76** | **48.54** | **17.72** | N/A | N/A | 34.18 |
| Multi-Domain. F: Flights, H: Hotels, R: Restaurants, A: Attractions | | | | | | | | | | | | |
| FH | 22.22 | 26.67 | 57.14 | 36.36 | 1.96 | 2.44 | 9.09 | 3.85 | 0.00 | N/A | N/A | 20.94 |
| FHA | 15.79 | 20.00 | 42.86 | 27.27 | 17.65 | 20.00 | 60.00 | 30.00 | 5.13 | N/A | N/A | 24.62 |
| FHR | 25.00 | 42.31 | 37.93 | 40.00 | 6.35 | 8.51 | 20.00 | 11.94 | 2.91 | N/A | N/A | 24.42 |
| HA | 16.67 | 24.44 | 34.38 | 28.57 | 9.09 | 10.39 | 42.11 | 16.67 | 4.13 | N/A | N/A | 29.36 |
| HR | 33.33 | 44.00 | 57.89 | 50.00 | 12.99 | 16.39 | 38.46 | 22.99 | 0.46 | N/A | N/A | 4.57 |

Table 17: Overall results using `OpenAI o4-mini` with few-shot in-context learning ($k = 13$).

| Domain | Tool Call | | | | Parameter Matching | | | | Code Exec. Rate | Information Seeking | | Cache Summary |
|---|---|---|---|---|---|---|---|---|---|---|---|---|
| | Acc. | Prec. | Rec. | F1 | Acc. | Prec. | Rec. | F1 | Acc. | SacreBLEU | BERTScore | EM |
| Single-Domain | | | | | | | | | | | | |
| Attractions | 86.39 | 94.19 | 91.25 | 92.70 | 72.36 | 83.18 | 84.76 | 83.96 | 68.07 | N/A | N/A | 82.35 |
| Flights | 89.66 | 94.79 | 94.30 | 94.55 | 71.08 | 81.10 | 85.20 | 83.10 | 79.27 | **59.27** | **91.02** | 71.95 |
| Hotels | 96.76 | 98.62 | 98.08 | 98.35 | 80.75 | 84.33 | **95.00** | 89.35 | **99.38** | 42.68 | 81.48 | 76.88 |
| Restaurants | **98.70** | **99.02** | **99.67** | **99.34** | **90.43** | **96.05** | 93.92 | **94.97** | 98.73 | 9.80 | 72.88 | **93.67** |
| Multi-Domain. F: Flights, H: Hotels, R: Restaurants, A: Attractions | | | | | | | | | | | | |
| FH | 81.93 | 90.69 | 89.45 | 90.07 | 62.80 | 72.56 | 82.37 | 77.15 | 89.74 | 13.31 | 81.60 | 64.10 |
| FHA | 83.46 | 91.94 | 90.05 | 90.98 | 74.47 | 86.48 | 84.28 | 85.37 | 95.38 | 39.21 | 88.14 | 69.74 |
| FHR | 77.85 | 87.88 | 87.22 | 87.55 | 73.71 | 88.99 | 81.10 | 84.86 | 97.09 | 17.98 | 82.25 | 70.93 |
| HA | 81.45 | 87.50 | 92.18 | 89.78 | 78.44 | 87.47 | 88.37 | 87.92 | 97.25 | 27.25 | 83.51 | 76.61 |
| HR | 77.85 | 83.57 | 91.91 | 87.54 | 71.08 | 80.26 | 86.14 | 83.09 | 95.89 | 25.43 | 87.48 | 57.99 |

Table 18: Overall results using `GPT-4.1` with few-shot in-context learning ($k = 13$).

| Domain | Tool Call | | | | Parameter Matching | | | | Code Exec. Rate | Information Seeking | | Cache Summary |
|---|---|---|---|---|---|---|---|---|---|---|---|---|
| | Acc. | Prec. | Rec. | F1 | Acc. | Prec. | Rec. | F1 | Acc. | SacreBLEU | BERTScore | EM |
| Single-Domain | | | | | | | | | | | | |
| Attractions | 59.28 | 73.72 | 75.16 | 74.43 | 36.48 | 42.96 | 70.73 | 53.46 | **75.63** | N/A | N/A | 59.66 |
| Flights | 80.42 | 88.12 | 90.21 | 89.15 | 55.31 | 63.46 | 81.15 | 71.22 | 60.98 | 26.68 | 82.58 | 58.54 |
| Hotels | 85.29 | 94.31 | 89.92 | 92.06 | 58.20 | 63.04 | 88.35 | 73.58 | 63.75 | 15.67 | 78.89 | 51.25 |
| Restaurants | **88.83** | **94.33** | 93.85 | **94.09** | 75.30 | 82.78 | 89.29 | 85.91 | 67.09 | 7.39 | 69.89 | **68.35** |
| Multi-Domain. F: Flights, H: Hotels, R: Restaurants, A: Attractions | | | | | | | | | | | | |
| FH | 68.49 | 78.44 | 84.37 | 81.30 | 42.73 | 52.34 | 69.95 | 59.88 | 64.53 | 10.63 | 79.05 | 47.86 |
| FHA | 67.57 | 74.67 | 87.67 | 80.65 | 50.57 | 61.81 | 73.55 | 67.17 | 57.95 | **30.03** | 83.18 | 48.21 |
| FHR | 68.33 | 75.38 | 87.95 | 81.18 | 59.25 | 72.39 | 76.55 | 74.41 | 61.63 | 9.97 | 77.20 | 55.23 |
| HA | 62.06 | 66.15 | 90.94 | 76.59 | 63.03 | 69.40 | 87.29 | 77.32 | 45.87 | 22.72 | 82.66 | 55.96 |
| HR | 60.05 | 66.21 | 86.59 | 75.04 | 50.57 | 57.30 | 81.15 | 67.17 | 48.40 | 18.10 | **84.31** | 29.68 |

Table 19: Overall results using `s1.1 32B` with few-shot in-context learning ($k = 13$) on `small` dataset.

| Domain | Tool Call | | | | Parameter Matching | | | | Code Exec. Rate | Information Seeking | | Cache Summary |
|---|---|---|---|---|---|---|---|---|---|---|---|---|
| | Acc. | Prec. | Rec. | F1 | Acc. | Prec. | Rec. | F1 | Acc. | SacreBLEU | BERTScore | EM |
| Single-Domain | | | | | | | | | | | | |
| Attractions | 40.00 | **100.00** | 40.00 | 57.14 | **66.67** | **100.00** | 66.67 | **80.00** | 10.08 | N/A | N/A | **40.34** |
| Flights | 9.80 | 10.42 | 62.50 | 17.86 | 8.62 | 8.93 | 71.43 | 15.87 | 8.54 | **30.91** | 71.39 | 37.80 |
| Hotels | 0.00 | 0.00 | 0.00 | 0.00 | 0.00 | 0.00 | 0.00 | 0.00 | 1.25 | N/A | N/A | 25.62 |
| Restaurants | 52.94 | 52.94 | **100.00** | 69.23 | 40.00 | 40.00 | **100.00** | 57.14 | **14.56** | 7.35 | 68.78 | 36.08 |
| Multi-Domain. F: Flights, H: Hotels, R: Restaurants, A: Attractions | | | | | | | | | | | | |
| FH | 18.87 | 27.78 | 37.04 | 31.75 | 16.28 | 25.93 | 30.43 | 28.00 | 0.85 | N/A | N/A | 20.94 |
| FHA | 9.23 | 9.52 | 75.00 | 16.90 | 30.00 | 37.50 | 60.00 | 46.15 | 5.64 | 353.69 | 56.04 | 24.62 |
| FHR | 9.84 | 13.64 | 26.09 | 17.91 | 12.90 | 80.00 | 13.33 | 22.86 | 4.65 | 647.95 | 56.10 | 24.42 |
| HA | **100.00** | **100.00** | **100.00** | **100.00** | 0.00 | 0.00 | 0.00 | 0.00 | 1.83 | 1232.03 | 79.41 | 30.28 |
| HR | 33.33 | **100.00** | 33.33 | 50.00 | 0.00 | 0.00 | 0.00 | 0.00 | 0.91 | 1585.06 | **84.21** | 4.57 |

Table 20: Overall results using `s1.1 7B` with few-shot in-context learning ($k = 13$) on `small` dataset.

| Domain | Tool Call | | | | Parameter Matching | | | | Code Exec. Rate | Information Seeking | | Cache Summary |
|---|---|---|---|---|---|---|---|---|---|---|---|---|
| | Acc. | Prec. | Rec. | F1 | Acc. | Prec. | Rec. | F1 | Acc. | SacreBLEU | BERTScore | EM |
| Single-Domain | | | | | | | | | | | | |
| Attractions | 32.43 | 42.86 | 57.14 | 48.98 | 20.83 | 23.81 | 62.50 | 34.48 | 14.29 | N/A | N/A | **39.50** |
| Flights | 7.69 | 14.29 | 14.29 | 14.29 | 0.00 | 0.00 | 0.00 | 0.00 | 6.10 | N/A | N/A | 37.20 |
| Hotels | **73.08** | **82.61** | **86.36** | **84.44** | 25.00 | 35.29 | 46.15 | 40.00 | 3.12 | N/A | N/A | 26.25 |
| Restaurants | 26.23 | 39.02 | 44.44 | 41.56 | **32.05** | **35.71** | **75.76** | **48.54** | **17.72** | N/A | N/A | 34.18 |
| Multi-Domain. F: Flights, H: Hotels, R: Restaurants, A: Attractions | | | | | | | | | | | | |
| FH | 22.22 | 26.67 | 57.14 | 36.36 | 1.96 | 2.44 | 9.09 | 3.85 | 0.00 | N/A | N/A | 20.94 |
| FHA | 15.79 | 20.00 | 42.86 | 27.27 | 17.65 | 20.00 | 60.00 | 30.00 | 5.13 | N/A | N/A | 24.62 |
| FHR | 25.00 | 42.31 | 37.93 | 40.00 | 6.35 | 8.51 | 20.00 | 11.94 | 2.91 | N/A | N/A | 24.42 |
| HA | 16.67 | 24.44 | 34.38 | 28.57 | 9.09 | 10.39 | 42.11 | 16.67 | 4.13 | N/A | N/A | 29.36 |
| HR | 33.33 | 44.00 | 57.89 | 50.00 | 12.99 | 16.39 | 38.46 | 22.99 | 0.46 | N/A | N/A | 4.57 |

Table 21: Overall results using `s1.1 3B` with few-shot in-context learning ($k = 13$) on `small` dataset.

| Domain | Tool Call | | | | Parameter Matching | | | | Code Exec. Rate | Information Seeking | | Cache Summary |
|---|---|---|---|---|---|---|---|---|---|---|---|---|
| | Acc. | Prec. | Rec. | F1 | Acc. | Prec. | Rec. | F1 | Acc. | SacreBLEU | BERTScore | EM |
| Single-Domain | | | | | | | | | | | | |
| Attractions | 0.00 | 0.00 | 0.00 | 0.00 | **0.00** | **0.00** | **0.00** | **0.00** | 9.24 | N/A | N/A | **40.34** |
| Flights | 0.00 | 0.00 | 0.00 | 0.00 | **0.00** | **0.00** | **0.00** | **0.00** | 6.10 | N/A | N/A | 37.20 |
| Hotels | 0.00 | 0.00 | 0.00 | 0.00 | **0.00** | **0.00** | **0.00** | **0.00** | 0.62 | N/A | N/A | 25.62 |
| Restaurants | 0.00 | 0.00 | 0.00 | 0.00 | **0.00** | **0.00** | **0.00** | **0.00** | **12.03** | N/A | N/A | 34.81 |
| Multi-Domain. F: Flights, H: Hotels, R: Restaurants, A: Attractions | | | | | | | | | | | | |
| FH | 0.00 | 0.00 | 0.00 | 0.00 | **0.00** | **0.00** | **0.00** | **0.00** | 0.00 | N/A | N/A | 20.94 |
| FHA | **53.33** | **53.33** | **100.00** | **69.57** | 0.00 | 0.00 | 0.00 | 0.00 | 5.64 | N/A | N/A | 24.62 |
| FHR | 0.00 | 0.00 | 0.00 | 0.00 | **0.00** | **0.00** | **0.00** | **0.00** | 0.58 | N/A | N/A | 24.42 |
| HA | 0.00 | 0.00 | 0.00 | 0.00 | **0.00** | **0.00** | **0.00** | **0.00** | 1.38 | N/A | N/A | 30.28 |
| HR | 0.00 | 0.00 | 0.00 | 0.00 | **0.00** | **0.00** | **0.00** | **0.00** | 0.00 | N/A | N/A | 4.57 |

Table 22: Overall results using `s1.1 1.5B` with few-shot in-context learning ($k = 13$) on `small` dataset.

| Domain | Tool Call | | | | Parameter Matching | | | | Code Exec. Rate | Information Seeking | | Cache Summary |
|---|---|---|---|---|---|---|---|---|---|---|---|---|
| | Acc. | Prec. | Rec. | F1 | Acc. | Prec. | Rec. | F1 | Acc. | SacreBLEU | BERTScore | EM |
| Single-Domain | | | | | | | | | | | | |
| Attractions | 37.85 | 63.21 | 48.55 | 54.92 | 25.00 | 67.50 | 28.42 | 40.00 | 66.39 | N/A | N/A | **52.94** |
| Flights | 35.76 | 66.67 | 43.54 | 52.68 | 15.38 | 52.50 | 17.67 | 26.67 | 73.17 | 13.84 | 81.46 | 39.02 |
| Hotels | **84.16** | **91.15** | **91.64** | **91.40** | **54.60** | 63.64 | **79.36** | **70.63** | 95.00 | **47.08** | 80.50 | 46.25 |
| Restaurants | 65.91 | 84.67 | 74.84 | 79.45 | 47.08 | 62.05 | 66.12 | 64.02 | 94.30 | 26.35 | 68.83 | 50.00 |
| Multi-Domain. F: Flights, H: Hotels, R: Restaurants, A: Attractions | | | | | | | | | | | | |
| FH | 58.54 | 88.54 | 63.34 | 73.85 | 31.36 | 60.00 | 39.65 | 47.75 | 83.76 | 28.66 | 83.25 | 32.48 |
| FHA | 39.13 | 64.12 | 50.10 | 56.25 | 33.33 | **68.25** | 39.45 | 50.00 | 83.08 | 16.01 | 81.02 | 33.33 |
| FHR | 47.84 | 71.76 | 58.94 | 64.72 | 33.04 | 55.32 | 45.07 | 49.67 | 90.12 | 35.20 | 84.51 | 39.53 |
| HA | 65.71 | 78.59 | 80.04 | 79.31 | 48.30 | 61.09 | 69.77 | 65.14 | **96.33** | 28.02 | 82.89 | 52.75 |
| HR | 53.53 | 70.37 | 69.10 | 69.73 | 35.72 | 51.48 | 53.86 | 52.64 | 95.89 | 22.93 | **84.85** | 21.92 |

Table 23: Overall results using `Phi-4-reasoning-plus` with few-shot in-context learning ($k = 13$) on `small` dataset.

| Domain | Tool Call | | | | Parameter Matching | | | | Code Exec. Rate | Information Seeking | | Cache Summary |
|---|---|---|---|---|---|---|---|---|---|---|---|---|
| | Acc. | Prec. | Rec. | F1 | Acc. | Prec. | Rec. | F1 | Acc. | SacreBLEU | BERTScore | EM |
| Single-Domain | | | | | | | | | | | | |
| Attractions | 28.76 | 44.44 | 44.90 | 44.67 | 12.61 | 24.59 | 20.55 | 22.39 | 38.66 | N/A | N/A | **44.54** |
| Flights | 26.19 | 31.88 | 59.46 | 41.51 | 16.07 | 17.85 | 61.73 | 27.69 | 24.39 | 14.61 | 82.50 | 34.76 |
| Hotels | 43.01 | 49.10 | 77.62 | 60.15 | 19.10 | 25.65 | 42.79 | 32.08 | 50.62 | **47.16** | 80.53 | 35.62 |
| Restaurants | 38.89 | 53.03 | 59.32 | 56.00 | 27.78 | 31.75 | **68.97** | 43.48 | **56.96** | 28.28 | 76.97 | 32.91 |
| Multi-Domain. F: Flights, H: Hotels, R: Restaurants, A: Attractions | | | | | | | | | | | | |
| FH | 33.30 | 42.52 | 60.56 | 49.96 | 17.47 | 21.15 | 50.10 | 29.75 | 41.03 | 26.02 | 82.63 | 19.66 |
| FHA | 29.22 | 37.52 | 56.90 | 45.22 | 17.44 | 22.17 | 45.01 | 29.70 | 30.77 | 14.73 | 79.65 | 24.62 |
| FHR | 28.43 | 33.84 | 63.98 | 44.27 | 18.11 | 21.20 | 55.44 | 30.67 | 32.56 | 30.97 | 83.02 | 28.49 |
| HA | 41.72 | 48.18 | 75.68 | 58.88 | 16.64 | 21.23 | 43.51 | 28.54 | 52.75 | 27.26 | 82.20 | 33.03 |
| HR | **53.33** | **61.18** | **80.60** | **69.56** | **29.81** | **36.79** | 61.10 | **45.92** | 53.42 | 27.95 | **83.77** | 11.42 |

Table 24: Overall results using `Llama 3.1 8B Instruct` with few-shot in-context learning ($k =$ 13) on `small` dataset.

| Domain | Tool Call | | | | Parameter Matching | | | | Code Exec. Rate | Information Seeking | | Cache Summary |
|---|---|---|---|---|---|---|---|---|---|---|---|---|
| | Acc. | Prec. | Rec. | F1 | Acc. | Prec. | Rec. | F1 | Acc. | SacreBLEU | BERTScore | EM |
| Single-Domain | | | | | | | | | | | | |
| Attractions | **91.08** | **97.98** | **92.82** | **95.33** | **71.67** | **90.53** | 77.48 | **83.50** | **99.16** | N/A | N/A | **79.83** |
| Flights | 83.52 | 91.35 | 90.69 | 91.02 | 57.44 | 66.67 | 80.58 | 72.96 | 86.59 | 40.20 | 86.39 | 64.63 |
| Hotels | 79.63 | 85.86 | 91.64 | 88.66 | 60.42 | 68.08 | 84.30 | 75.32 | 72.50 | 48.25 | 80.74 | 71.25 |
| Restaurants | 73.07 | 80.83 | 88.39 | 84.44 | 56.64 | 66.51 | 79.23 | 72.32 | 96.20 | **95.72** | **98.82** | 63.92 |
| Multi-Domain. F: Flights, H: Hotels, R: Restaurants, A: Attractions | | | | | | | | | | | | |
| FH | 79.81 | 91.89 | 85.86 | 88.77 | 57.57 | 75.62 | 70.68 | 73.07 | 88.03 | 24.26 | 80.38 | 57.69 |
| FHA | 78.97 | 86.90 | 89.64 | 88.25 | 62.55 | 74.90 | 79.14 | 76.96 | 80.00 | 23.37 | 83.95 | 55.90 |
| FHR | 69.43 | 84.01 | 80.00 | 81.95 | 53.50 | 70.00 | 69.42 | 69.71 | 81.40 | 47.69 | 84.77 | 54.07 |
| HA | 61.03 | 72.63 | 79.27 | 75.80 | 58.57 | 68.65 | 79.95 | 73.87 | 69.27 | 29.67 | 83.19 | 64.68 |
| HR | 80.83 | 90.41 | 88.41 | 89.40 | 67.75 | 77.48 | **84.36** | 80.78 | 80.82 | 29.94 | 87.88 | 62.10 |

Table 25: Overall results using `Llama 3.1 8B Instruct SFT` with few-shot in-context learning ($k = 13$) on `small` dataset.

| Domain | Tool Call | | | | Parameter Matching | | | | Code Exec. Rate | Information Seeking | | Cache Summary |
|---|---|---|---|---|---|---|---|---|---|---|---|---|
| | Acc. | Prec. | Rec. | F1 | Acc. | Prec. | Rec. | F1 | Acc. | SacreBLEU | BERTScore | EM |
| Single-Domain | | | | | | | | | | | | |
| Attractions | **91.08** | **97.98** | **92.82** | **95.33** | 71.67 | 90.53 | 77.48 | 83.50 | **99.16** | N/A | N/A | 79.83 |
| Attractions | 77.10 | 88.60 | 85.59 | 87.07 | **74.74** | **93.42** | 78.89 | **85.54** | 53.78 | N/A | N/A | 73.95 |
| Flights | 72.80 | 81.29 | 87.46 | 84.26 | 52.25 | 59.52 | 81.07 | 68.64 | 70.73 | 21.50 | 82.30 | 52.44 |
| Hotels | 88.29 | **93.93** | 93.63 | 93.78 | 72.03 | 80.53 | 87.22 | 83.74 | 82.50 | 45.83 | 80.23 | 64.38 |
| Restaurants | 89.42 | 92.38 | **96.54** | **94.42** | 72.30 | 80.63 | **87.50** | 83.92 | 88.61 | **53.50** | 84.86 | **82.91** |
| Multi-Domain. F: Flights, H: Hotels, R: Restaurants, A: Attractions | | | | | | | | | | | | |
| FH | 59.47 | 68.55 | 81.79 | 74.59 | 34.27 | 39.85 | 71.01 | 51.05 | 75.64 | 26.27 | 82.61 | 33.76 |
| FHA | 73.44 | 78.57 | 91.84 | 84.69 | 59.36 | 65.19 | 86.92 | 74.50 | 79.49 | 18.17 | 81.41 | 45.13 |
| FHR | 61.48 | 70.32 | 83.01 | 76.14 | 55.79 | 65.05 | 79.68 | 71.62 | 78.49 | 42.93 | **86.68** | 43.60 |
| HA | 64.54 | 72.65 | 85.25 | 78.45 | 55.17 | 64.76 | 78.83 | 71.11 | 78.90 | 29.06 | 83.20 | 61.93 |
| HR | 64.01 | 74.20 | 82.34 | 78.06 | 56.51 | 70.18 | 74.38 | 72.22 | **90.41** | 22.39 | 84.33 | 27.40 |

Table 26: Overall results using `Llama 3.1 70B Instruct` with few-shot in-context learning ($k =$ 13) on `small` dataset.

| Domain | Tool Call | | | | Parameter Matching | | | | Code Exec. Rate | Information Seeking | | Cache Summary |
| --- | --- | --- | --- | --- | --- | --- | --- | --- | --- | --- | --- | --- |
| | Acc. | Prec. | Rec. | F1 | Acc. | Prec. | Rec. | F1 | Acc. | SacreBLEU | BERTScore | EM |
| Single-Domain | | | | | | | | | | | | |
| Attractions | **91.08** | **97.98** | **92.82** | **95.33** | **71.67** | **90.53** | 77.48 | **83.50** | **99.16** | N/A | N/A | **79.83** |
| Attractions | 76.73 | 90.37 | 83.56 | 86.83 | 70.00 | 89.53 | 76.24 | 82.35 | 62.18 | N/A | N/A | 77.31 |
| Flights | 56.86 | 75.86 | 69.42 | 72.50 | 29.81 | 36.61 | 61.62 | 45.93 | 83.54 | 20.11 | 82.87 | 50.61 |
| Hotels | **92.95** | **95.96** | **96.74** | **96.35** | 74.94 | 80.99 | 90.94 | 85.67 | 97.50 | **46.98** | 80.32 | 73.75 |
| Restaurants | 92.57 | 95.83 | 96.45 | 96.14 | **84.77** | **92.27** | **91.26** | **91.76** | 99.37 | 41.56 | 80.99 | **91.14** |
| Multi-Domain. F: Flights, H: Hotels, R: Restaurants, A: Attractions | | | | | | | | | | | | |
| FH | 42.34 | 53.03 | 67.74 | 59.49 | 21.39 | 24.89 | 60.33 | 35.24 | 96.15 | 27.79 | 83.65 | 37.61 |
| FHA | 63.00 | 72.56 | 82.70 | 77.30 | 45.53 | 52.91 | 76.54 | 62.57 | 94.36 | 20.28 | 82.16 | 45.13 |
| FHR | 53.69 | 62.20 | 79.71 | 69.87 | 44.83 | 53.28 | 73.88 | 61.91 | 91.28 | 40.87 | **85.90** | 44.19 |
| HA | 66.62 | 70.58 | 92.23 | 79.96 | 64.59 | 72.49 | 85.57 | 78.49 | 98.62 | 30.43 | 82.82 | 57.34 |
| HR | 58.44 | 69.38 | 78.76 | 73.77 | 46.95 | 56.04 | 74.32 | 63.90 | **100.00** | 23.04 | 84.22 | 37.44 |

Table 27: Overall results using `Llama 3.3 70B Instruct` with few-shot in-context learning ($k = 13$) on `small` dataset.

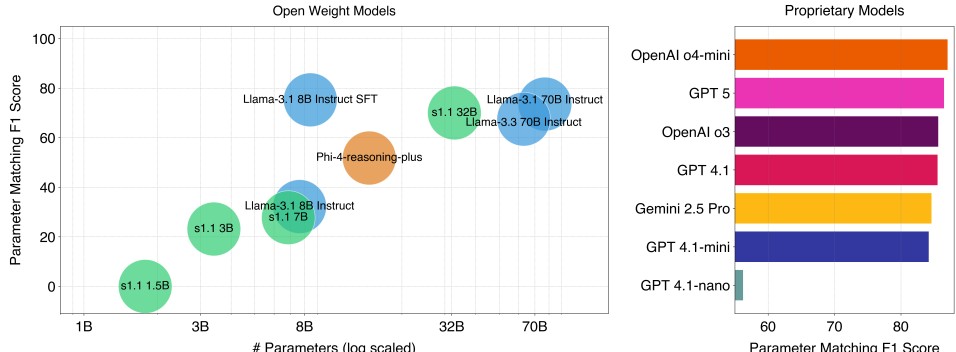

Figure 7: Parameter Matching F1 score between open weight and proprietary models with few-shot in-context learning ($k$=13).

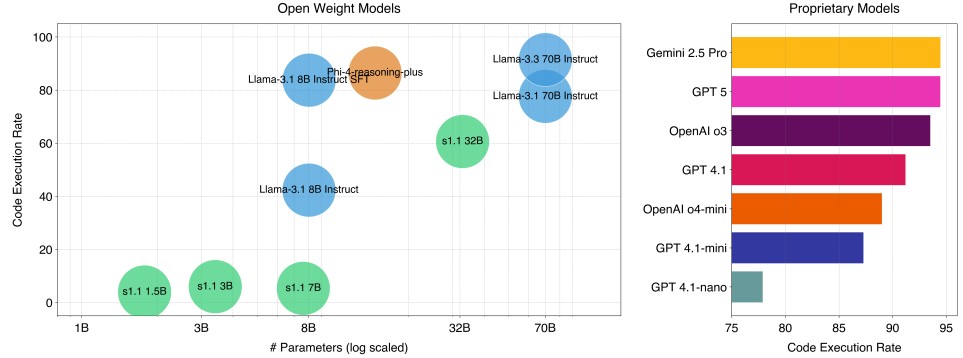

Figure 8: Code Execution Rate Accuracy between open weight and proprietary models with few-shot in-context learning ($k$=13).

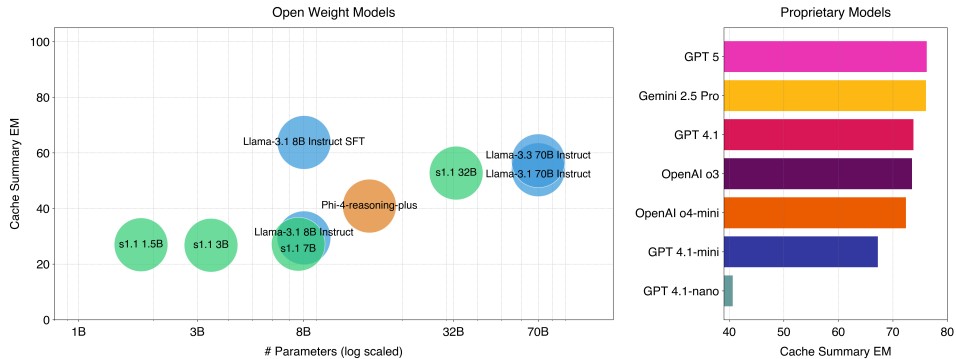

Figure 9: Cache Summary Exact Match between open weight and proprietary models with few-shot in-context learning ($k$=13).

# D Error Analysis on large dataset results

Upon closer examination of the errors observed during inference on the `large` dataset, we categorize them into four main types:

- **Validation Error**: Errors related to improper arguments being passed to the defined tools.
- **Variable Not Defined**: Error indicating code uses an underfined variable.
- **Index Out Of Range**: Error indicating generated code involves an index out of range, usually corresponding to the list data structure in Python.
- **Other**: Other errors present within the generated code.

Overall, we notice that validation errors are consistently present within the results of all of the models we evaluated.

## D.1 Llama 3.3 70B Instruct

The `Llama 3.3 70B Instruct` model contains the fewest number of overall errors among the five models evaluated with a total of 1,466. This supports the idea that the larger number of parameters helps the model better understand the task as well passing in the proper parameters to the tools to generate high quality code.

| Domain | # Turns | # Validation Error | # Variable Not Defined | # Index Out Of Range | # Other |
|---|---|---|---|---|---|
| Single-Domain | | | | | |
| Flights | 8,200 | 518 | 2 | 30 | 0 |
| Hotels | 8,000 | 41 | 0 | 0 | 0 |
| Restaurants | 7,900 | 29 | 0 | 0 | 0 |
| Attractions | 5,975 | 4 | 2 | 0 | 1 |
| Multi-Domain. F: Flights, H: Hotels, R: Restaurants, A: Attractions | | | | | |
| F-H | 11,700 | 158 | 28 | 0 | 0 |
| H-R | 10,950 | 14 | 2 | 0 | 1 |
| H-A | 11,000 | 48 | 33 | 0 | 48 |
| F-H-A | 9,775 | 148 | 6 | 7 | 3 |
| F-H-R | 8,575 | 330 | 5 | 1 | 7 |
| Total | - | 1290 | 78 | 38 | 60 |

Table 28: Overall error analysis results using `Llama 3.3 70B Instruct`.

## D.2 Llama 3.1 8B Instruct

The `Llama 3.1 8B Instruct` model consists of the most number of total errors among all 4 categories which were tracked. The only category it performs well in was the `Index Out Of Range` error where there are only 5 instances.

| Domain | # Turns | # Validation Error | # Variable Not Defined | # Index Out Of Range | # Other |
|--------|---------|--------------------|------------------------|----------------------|---------|
| Single-Domain | | | | | |
| Flights | 8,200 | 737 | 830 | 0 | 468 |
| Hotels | 8,000 | 417 | 218 | 0 | 820 |
| Restaurants | 7,900 | 513 | 14 | 1 | 183 |
| Attractions | 5,975 | 507 | 1 | 0 | 243 |
| Multi-Domain. F: Flights, H: Hotels, R: Restaurants, A: Attractions | | | | | |
| F-H | 11,700 | 834 | 1087 | 0 | 408 |
| H-R | 10,950 | 1532 | 18 | 1 | 522 |
| H-A | 11,000 | 1024 | 167 | 0 | 308 |
| F-H-A | 9,775 | 1163 | 990 | 0 | 575 |
| F-H-R | 8,575 | 812 | 761 | 3 | 495 |
| Total | - | 7539 | 4086 | 5 | 4022 |

Table 29: Overall error analysis results using Llama 3.1 8B Instruct.

## D.3    Llama 3.1 8B Instruct SFT

The Llama 3.1 8B Instruct SFT does drastically improve on the number of Validation, Variable Not Defined and Other categorized errors compared to the original model. This indicates that the model is able to better learn about calling the tools and writing higher quality code than the original model.

| Domain | # Turns | # Validation Error | # Variable Not Defined | # Index Out Of Range | # Other |
|--------|---------|--------------------|------------------------|----------------------|---------|
| Single-Domain | | | | | |
| Flights | 8,200 | 408 | 176 | 42 | 26 |
| Hotels | 8,000 | 1,099 | 16 | 0 | 0 |
| Restaurants | 7,900 | 209 | 5 | 0 | 2 |
| Attractions | 5,975 | 18 | 9 | 0 | 8 |
| Multi-Domain. F: Flights, H: Hotels, R: Restaurants, A: Attractions | | | | | |
| F-H | 11,700 | 454 | 149 | 0 | 22 |
| H-R | 10,950 | 692 | 185 | 0 | 17 |
| H-A | 11,000 | 1248 | 315 | 0 | 1 |
| F-H-A | 9,775 | 789 | 124 | 9 | 27 |
| F-H-R | 8,575 | 541 | 109 | 0 | 101 |
| Total | - | 5458 | 1088 | 51 | 204 |

Table 30: Overall error analysis results using Llama 3.1 8B SFT Instruct.

### D.4   s1.1 32B

Overall, the `s1.1 32B` model performs the best when it comes to `Validation` errors with a total of 937 errors. However, it also consists of the highest amount of `Index Out of Range` errors and the 2nd highest amount of `Variable Not Defined` errors.

| Domain | # Turns | # Validation Error | # Variable Not Defined | # Index Out Of Range | # Other |
|---|---|---|---|---|---|
| `Single-Domain` | | | | | |
| Flights | 8,200 | 559 | 16 | 3 | 40 |
| Hotels | 8,000 | 92 | 1 | 39 | 27 |
| Restaurants | 7,900 | 20 | 7 | 0 | 28 |
| Attractions | 5,975 | 11 | 27 | 0 | 76 |
| `Multi-Domain`. F: Flights, H: Hotels, R: Restaurants, A: Attractions | | | | | |
| F-H | 11,700 | 24 | 34 | 114 | 63 |
| H-R | 10,950 | 54 | 895 | 102 | 34 |
| H-A | 11,000 | 3 | 1071 | 43 | 23 |
| F-H-A | 9,775 | 89 | 579 | 25 | 49 |
| F-H-R | 8,575 | 85 | 241 | 7 | 28 |
| `Total` | - | 937 | 2871 | 333 | 368 |

Table 31: Overall error analysis results using `s1.1`.

### D.5   Phi-4-reasoning-plus

The `Phi-4-reasoning-plus` has 1,498 validation errors which is the third most among the 5 models. The model does perform reasonably well with the other error categories and has the fewest number of `Index Out Of Range` and `Variable Not Defined` errors with 1 and 16 respectively.

| Domain | # Turns | # Validation Error | # Variable Not Defined | # Index Out Of Range | # Other |
|---|---|---|---|---|---|
| `Single-Domain` | | | | | |
| Flights | 8,200 | 103 | 7 | 1 | 1 |
| Hotels | 8,000 | 277 | 0 | 0 | 1 |
| Restaurants | 7,900 | 259 | 0 | 0 | 19 |
| Attractions | 5,975 | 0 | 0 | 0 | 0 |
| `Multi-Domain`. F: Flights, H: Hotels, R: Restaurants, A: Attractions | | | | | |
| F-H | 11,700 | 212 | 2 | 0 | 7 |
| H-R | 10,950 | 212 | 0 | 0 | 1 |
| H-A | 11,000 | 162 | 0 | 0 | 6 |
| F-H-A | 9,775 | 120 | 3 | 0 | 5 |
| F-H-R | 8,575 | 153 | 4 | 0 | 32 |
| `Total` | - | 1498 | 16 | 1 | 72 |

Table 32: Overall error analysis results using `Phi-4-reasoning-plus`.

# E   Model and Prompt Configuration

We evaluate a diverse set of models on two dataset splits: a `small` dataset and a `large` dataset.

- On the **large dataset**, we evaluate the open-weight models: `Phi-4-reasoning-plus`, `s1.1 32B`, `Llama 3.1 8B Instruct`, `Llama 3.3 70B Instruct`, and a fine-tuned `Llama 3.1 8B Instruct SFT` (detailed below).

- On the **small dataset**, we evaluate a broader set including proprietary models: `Phi-4-reasoning-plus`, `s1.1 1.5B`, `s1.1 3B`, `s1.1 7B`, `s1.1 32B`, `Llama 3.1 8B Instruct`, `Llama 3.1 8B Instruct SFT`, `Llama 3.1 70B Instruct`, `Llama 3.3 70B Instruct`, `Gemini 2.5 Pro`, `OpenAI o3`, `OpenAI o4-mini`, `GPT 4.1`, `GPT 4.1-mini`, `GPT 4.1-nano`, and `GPT 5`.

All model evaluations are conducted in a *few-shot* setting, using $k = 13$ example turns included in the prompt to guide model behavior. The full prompts used for inference are provided in Appendix L and Appendix M. For decoding, we use a temperature of 0.1, top-k sampling of 10, and a maximum new token limit of 4,000. We host our open-weight models using vLLM on a node with 8x A100 40GB GPUs.

To evaluate the effectiveness of domain adaptation, we fine-tune `Llama 3.1 8B Instruct` on the T1 dataset. The following script, which uses the TRL library (v0.17.0), can be used to replicate this Supervised Fine-Tuning (SFT):

```
>> git clone https://github.com/huggingface/trl.git
>> cd trl
>> git checkout v0.17.0 # We use TRL version 0.17.0
>> pip install -e .
>> accelerate launch --config_file trl/examples/accelerate_configs/multi_gpu.yaml \
    trl/scripts/sft.py \
    --model_name_or_path <YOUR PATH TO A Meta-Llama-3.1-8B-Instruct LOCALLY> \
    --dataset_name <TRAIN SET OF THE TOOL CALLING IN HUGGINGFACE DATASET FORMAT> \
    --learning_rate 2.0e-5 \
    --num_train_epochs 1 \
    --packing \
    --per_device_train_batch_size 1 \
    --gradient_accumulation_steps 8 \
    --gradient_checkpointing \
    --logging_steps 25 \
    --eval_strategy steps \
    --eval_steps 100 \
    --use_peft \
    --lora_r 32 \
    --lora_alpha 32 \
    --lora_target_modules q_proj k_proj v_proj o_proj gate_proj up_proj down_proj \
    --output_dir <SOME OUTPUT DIRECTORY> \
    --report_to tensorboard \
    --torch_dtype bfloat16
```

To assess output quality and stability, we computed performance metrics for `Llama 3.3 70B Instruct` and `Gemini 2.5 Pro` on the **small dataset** across three different random seeds. Both models exhibited consistent performance, with standard deviations below one percentage point, indicating high stability. Table 33 shows the results for Tool Call Parameter Matching and Code Execution Rate metrics. Table 34 shows the results for Information Seeking metrics.

# F   More Information on T1 Dataset

Table 35 shows the attributes for each domain. Table 36 shows the tools used for each domain.

| Model | Task | Acc. | Prec. | Rec. | F1 |
|---|---|---|---|---|---|
| Gemini 2.5 Pro | Tool Call Parameter Matching | $88.93 \pm 0.35$ | $93.26 \pm 0.20$ | $94.92 \pm 0.26$ | $94.04 \pm 0.21$ |
| | Code Execution Rate | $73.07 \pm 0.54$ | $81.97 \pm 0.45$ | $86.69 \pm 0.33$ | $84.19 \pm 0.39$ |
| Llama-3.3 70B Instruct | Tool Call Parameter Matching | $66.98 \pm 0.12$ | $76.51 \pm 0.38$ | $82.77 \pm 0.42$ | $79.15 \pm 0.04$ |
| | Code Execution Rate | $53.40 \pm 0.22$ | $61.91 \pm 0.39$ | $76.81 \pm 0.06$ | $67.48 \pm 0.20$ |

Table 33: Model performance on Tool Call Parameter Matching and Code Execution Rate tasks cross different seeds.

| Model | Acc. | SacreBLEU | BERTScore | EM |
|---|---|---|---|---|
| Gemini 2.5 Pro | $94.23 \pm 0.24$ | $7.56 \pm 0.40$ | $74.54 \pm 0.13$ | $75.44 \pm 0.77$ |
| Llama-3.3 70B Instruct | $91.65 \pm 0.64$ | $31.04 \pm 0.30$ | $82.73 \pm 0.12$ | $57.55 \pm 0.33$ |

Table 34: Model performance on the Information Seeking task cross different seeds.

| Domain | Attributes |
|---|---|
| Flight | airline, flight_id, start_airport, start_airport_latitude, start_airport_longitude, start_airport_code, end_airport, end_airport_latitude, end_airport_longitude, end_airport_code, economy_class_option_present, business_class_option_present, first_class_option_present, distance_miles, duration_minutes, departure_time, arrival_time, number_of_layovers, first_layover_airport, |
| Hotel | hotel_name, city, state, neighborhood, latitude, longitude, rating, stars, max_occupancy, gym_present, pool_present, price_per_night, num_rooms_available, breakfast_included, smoking_allowed, air_conditioning_present, heating_present, free_wifi_included, airport_shuttle_present, is_pet_friendly, has_spa_services, has_room_service, has_beach_access, has_fitness_class, has_laundry_service, has_valet_parking, has_balcony, has_rooftop_bar, has_inroom_Kitchen, has_kids_club, has_meeting_rooms, has_electric_vehicle_charging, has_hot_tub, has_sauna, has_free_parking, is_wheelchair_accessible, has_skiing_lodging, has_ocean_view_rooms_present, has_city_view_rooms_present, start_date_available, end_date_available |
| Attraction | city, state, name, type, latitude, longitude, neighborhood |
| Restaurant | restaurant_name, city, state, neighborhood, latitude, longitude, rating, price, price_per_person, has_nut_allergy_options, has_dairy_allergy_options, has_shell_fish_allergy_options, has_tomato_allergy_options, has_nightshade_allergy_options, has_gluten_free_options, has_vegetarian_options, has_vegan_options, has_kosher_options, has_halal_options, cuisine |

Table 35: Domains and attributes.

| Domain | Tools |
|---|---|
| **Flights** | search_flights, filter_flights |
| **Hotels** | search_hotels, filter_hotels |
| **Restaurants** | search_restaurants, filter_restaurants |
| **Attractions** | search_attractions, filter_attractions |
| **Multi-domain** | search_nearest |
| **Common** | save_to_cache, get_results_from_cache, sort_results, seek_information, adjust_date |

Table 36: List of tools.

# G    Sample Conversation from T1 Dataset

| Attribute | Type | Description |
|---|---|---|
| airline | string | Airline of the flight, |
| flight_classes | string | Classes for the flight (economy, business, first) |
| num_layovers | integer | Number of layovers for the flight, between 0 and 2 |
| layover_duration_amount | integer | Duration of a layover flight. Between 1 and 6 hours |
| airports | list | Airport information for each flight including the city, state, airport code and airport name. |

Table 37: Flight ontology attributes.

| Attribute | Type | Description |
|---|---|---|
| city | string | City that the hotel is in |
| state | string | State that the hotel is in |
| neighborhood | string | Neighborhood within the city that the hotel is in |
| stars | integer | Star rating of the hotel between 1 and 5 |
| rating | float | Customer rating of the hotel between 1.0 and 5.0, incremented by 0.1 |
| price_per_night | integer | Price per night of the hotel ranging between 20 to 2000 dollars |
| max_occupancy | integer | Maximum occupancy per room ranging between 1 and 7 |
| gym_present | boolean | Whether or not the hotel has a gym |
| pool_present | boolean | Whether or not the hotel has a pool |
| breakfast_included | boolean | Whether or not the hotel has breakfast included |
| smoking_allowed | boolean | Whether or not the hotel allows smoking |
| air_conditioning_present | boolean | Whether or not the hotel has air conditioning |
| heating_present | boolean | Whether or not the hotel has heating |
| free_wifi_included | boolean | Whether or not the hotel includes free WiFi |
| airport_shuttle_present | boolean | Whether or not the hotel has an airport shuffle |
| is_pet_friendly | boolean | Whether or not the hotel allows pets |
| has_spa_services | boolean | Whether or not the hotel has spa services |
| has_room_service | boolean | Whether or not the hotel has room service |
| has_beach_access | boolean | Whether or not the hotel has access to a beach |
| has_business_center | boolean | Whether or not the hotel has a business center |
| has_fitness_classes | boolean | Whether or not the hotel has fitness classes |
| has_laundry_service | boolean | Whether or not the hotel has laundry services |
| has_valet_parking | boolean | Whether or not the hotel has valet parking |
| has_balcony | boolean | Whether or not the hotel has a balcony |
| has_rooftop_bar | boolean | Whether or not the hotel has a rooftop bar |
| has_inroom_kitchen | boolean | Whether or not the hotel has an inroom kitchen |
| has_kids_club | boolean | Whether or not the hotel has a kids club |
| has_meeting_rooms | boolean | Whether or not the hotel has meeting rooms |
| has_electric_vehicle_charging | boolean | Whether or not the hotel has electric vehicle charging |
| has_hot_tub | boolean | Whether or not the hotel has a hot tub |
| has_sauna | boolean | Whether or not the hotel has a sauna |
| has_free_parking | boolean | Whether or not the hotel has free parking |
| is_wheelchair_accessible | boolean | Whether or not the hotel is wheelchair accessible |
| has_skiing_lodging | boolean | Whether or not the hotel has skiing and lodging |
| ocean_view_rooms_present | boolean | Whether or not the hotel has rooms with a view of the ocean |
| city_view_rooms_present | boolean | Whether or not the hotel has rooms with views of the city |

Table 38: Hotel ontology attributes.

| Attribute | Type | Description |
|---|---|---|
| city | string | City that the restaurant is in |
| state | string | State that the restaurant is in |
| neighborhood | string | Neighborhood within the city that the restaurant is in |
| rating | float | Customer rating of the restaurant between 1.0 and 5.0, incremented by 0.1 |
| price_per_person | integer | Average price per person at the restaurant |
| cuisine | string | Cuisine of the restaurant |
| has_nut_allergy_options | boolean | Whether or not the restaurant has any options for individuals with an allergy to nuts |
| has_dairy_allergy_options | boolean | Whether or not restaurant has any options for individuals with an allergy to dairy products |
| has_shell_fish_allergy_options | boolean | Whether or not the restaurant has any options for individuals with an allergy to shell fish |
| has_tomato_allergy_options | boolean | Whether or not the restaurant has options for individuals with an allergy to tomatoes |
| has_nightshade_allergy_options | boolean | Whether or not the restaurant has options for individuals with an allergy to nightshades |
| has_gluten_free_options | boolean | Whether or not the restaurant has gluten free options |
| has_vegetarian_options | boolean | Whether or not the restaurant has vegetarian options |
| has_vegan_options | boolean | Whether or not the restaurant has vegan options |
| has_halal_options | boolean | Whether or not the restaurant has halal options |
| has_kosher_options | boolean | Whether or not the hotel has kosher options |

Table 39: Restaurant ontology attributes.

| Attribute | Type | Description |
|---|---|---|
| city | string | City that the attraction is in |
| state | string | State that the attraction is in |
| neighborhood | string | Neighborhood within the city that the attraction is in |
| type | string | Type of the attraction |

Table 40: Attraction ontology attributes.

## Box 1. Sample conversation for attractions

**assistant**: Hello! Are you looking for something to do in your free time?

**user**: Yeah, I am thinking of visiting some scenic attractions in San Antonio.

**assistant**: San Antonio has a lot of great Scenic spots. Have you considered Downtown San Antonio?

**user**: Actually, I haven't. What's there?

**assistant**: It's a great area with a lot of Scenic attractions. I can give you some recommendations.

**user**: Okay, that sounds good.

## Box 2. Sample conversation for flights

**assistant**: Welcome! What can I help you with today?

**user**: I need to book a flight from Colorado Springs to San Antonio for my anniversary, and I'm looking for something special.

**assistant**: Happy anniversary! What date are you planning to travel?

**user**: We're thinking of May 21, 2025.

**assistant**: That's great! What kind of flight experience are you looking for? Luxury, perhaps?

**user**: Yes, something luxurious. And we'd like to depart from COS.

**assistant**: I'll look into that. What's your budget for the flight?

**user**: We're looking to spend around 1100.

## Box 3. Sample conversation for hotels

**assistant**: Hello, what brings you here today?

**user**: I'm looking for a hotel in St. Louis for my anniversary, can you help me?

**assistant**: Congratulations on your anniversary! What are your check-in and check-out dates?

**user**: We're planning to check-in on May 18, 2025 and check-out on May 29, 2025.

**assistant**: That's great! How many rooms do you need and what's your budget per night?

**user**: We need 3 room and our budget is around 550 per night.

**assistant**: I see. What kind of amenities are you looking for in a hotel?

**user**: We'd like a hotel with a gym, pool, and air conditioning.

**assistant**: Got it. Would you prefer a hotel in a specific neighborhood in St. Louis?

user: Yes, we'd like to stay in Downtown St. Louis.

---

**Box 4. Sample conversation for restaurants**

**assistant**: Welcome, what brings you here today?

**user**: I'm gluten-free and I'm looking for a restaurant that can accommodate my dietary needs.

**assistant**: Don't worry, I'm here to help. What kind of cuisine are you interested in?

**user**: I'm really craving Livonian.

**assistant**: There are some great Livonian places that offer gluten-free options. Are you looking for something in a specific neighborhood?

**user**: Yeah, I'd love something in Las Vegas somewhere in Summerlin.

**assistant**: I can check what's available in Summerlin. What's your budget for this meal?

**user**: I'm looking to spend around 120 per person.

**assistant**: That helps. Would you like me to recommend a few places that fit your criteria?

**user**: Please do.

---

**Box 5. Sample conversation for flight and hotels**

**assistant**: Hello, how can I assist you today?

**user**: I'm looking for a flight from Atlanta to Memphis and a hotel in Memphis

**assistant**: What are your travel dates?

**user**: My departure date is May 18, 2025 and I'd like to return on June 1, 2025

**assistant**: What class of flight are you interested in?

**user**: I'd prefer business

**assistant**: And for your hotel, what star rating are you looking for?

**user**: I'd like a 2 star hotel in Downtown Memphis

---

**Box 6. Sample conversation for hotel and restaurant**

**assistant**: Hello! How can I assist you today?

**user**: I'm looking for a hotel in San Antonio and a restaurant that serves Turkish.

**assistant**: That sounds like a great plan! Can you tell me a bit more about your hotel preferences, such as the number of rooms and check-in date?

**user**: I need 3 rooms for 5 people, checking in on May 22, 2025 and checking out on May 27, 2025.

**assistant**: I'd be happy to help you with that. What's your budget per night for the hotel?

**user**: I'm looking to spend around 50 per night.

**assistant**: That's helpful to know. For the restaurant, are you looking for something with a specific rating or price range?

**user**: Yes, I'd like a restaurant with a rating of at least 3.5 and a price range of around 85.

**assistant**: I'll keep that in mind. Would you like the hotel to have any specific amenities, such as a gym or pool?

**user**: Yes, a pool would be great. And can you recommend a restaurant with Turkish near the hotel?

---

**Box 7. Sample conversation for hotel and attraction**

**assistant**: Hi there! What brings you here today?

**user**: I'm planning a last-minute trip to Portland for 2 people.

**assistant**: Last-minute trips can be exciting! What's your budget for the hotel?

**user**: We're looking to spend around 55 per night.

**assistant**: Okay, that helps. What's your check-in and check-out date?

**user**: We're checking in on June 28, 2025 and checking out on July 10, 2025.

**assistant**: Are you interested in any specific attractions in Portland?

**user**: Yes, we want to visit Pittock Mansion, which is a Social attraction.

**assistant**: That sounds like a great choice! I can help you find a hotel near Pittock Mansion. What kind of amenities are you look for in the hotel?

**user**: We need a hotel with a gym and wifi.

---

**Box 8. Sample conversation for flight-hotel-attraction**

**assistant**: Hello, I'd be happy to help you plan your trip. What's on your mind?

**user**: I'm thinking of going to Las Vegas for my birthday and I need a business flight from Tucson.

**assistant**: Happy birthday in advance. When are you thinking of traveling?

**user**: We're looking at May 19, 2025 to May 27, 2025.

**assistant**: That's a great time. How many people are traveling?

**user**: Just 2 of us.

**assistant**: Lovely. For your stay, what kind of hotel are you envisioning?

**user**: A 2 star hotel with a spa and room service would be perfect. And we'd love to stay near Fountains of Bellagio with a beautiful view.

**assistant**: That sounds like a great choice! I can help you find a hotel near Pittock Mansion. What kind of amenities are you look for in the hotel?

---

**Box 9. Sample conversation for flight-hotel-restaurant**

**assistant**: Welcome to our travel assistant. How can I help you today?

**user**: I'm looking to travel from Tucson to St. Louis for a business trip.

**assistant**: What are your preferred travel dates?

**user**: I'd like to fly out on May 18, 2025 and come back on June 1, 2025

**assistant**: What class would you like to fly in?

**user**: I prefer first.

**assistant**: Great, I can look into that. Where would you like to stay in St. Louis?

**user**: I'd like a hotel in Downtown St. Louis with a 3 star rating.

**assistant**: And what kind of restaurant are you looking for?

**user**: I'd like a restaurant in Downtown St. Louis with a rating of at least 3.5.

---

## H  System Prompt Design for Single-Domain Template Generation

This section details the unified system prompt and its domain-specific customizations used for generating diverse, realistic multi-turn conversations for four domains: **Flights, Hotels, Attractions, and Restaurants**.

### H.1  Unified Prompt Instructions (Common Across Domains)

The following design principles apply across all domains to ensure consistency, diversity, and realism in the generated templates:

- **Conversation Format:**
  - Each template contains 6–8 dialogue turns (3–4 exchanges).
  - Each template starts with an `assistant` turn and ends with a `user` turn.
  - Turns alternate between `assistant:` and `user:`.

- **Placeholder Handling:**
  - Use numbered placeholders like `<CITY_1>`, `<PRICE_2>`.
  - Numbering resets at the start of each new template.

- **Assistant Behavior Diversity:**
  - Vary personality: formal/informal, concise/verbose.
  - Ask both open-ended and specific questions.
  - Do not assume user intent or provide options in the first message.

- **– Correct misunderstandings if needed.**
- **User Behavior Diversity:**
  - – Include verbose, terse, indecisive, and overly specific users.
  - – Some users provide minimal context; others give excess information.
  - – Include questions that revisit or skip steps.
- **Flow Diversity:**
  - – Avoid fixed dialogue patterns (e.g., greet → date → options).
  - – Allow nonlinear flow: backtracking, parallel requests, etc.
- **Language Diversity:**
  - – Vary greetings, transitions, sentence structure, and phrasings.
  - – No repeated phrasing across templates.
- **Output Requirements:**
  - – Output format is strict JSON with keys like `"template_1"` through `"template_30"`.
  - – Each value is a newline-separated conversation string.
  - – No preambles or postambles allowed.

## H.2 Domain-Specific Customization

Figure 41 presents the domain-specific customization we use for T1.

| Domain | Placeholders Used | Required Scenarios |
|---|---|---|
| **Flight** | `<CITY_x>`, `<CLASS_x>`, `<DEPARTURE_DATE_x>`, `<DEPARTURE_TIME_x>`, `<ARRIVAL_DATE_x>`, `<ARRIVAL_TIME_x>`, `<AIRLINE_x>`, `<PRICE_x>`, `<AIRPORT_x>`, `<NUM_TRAVELERS_x>` | Round trip (with 2 departure dates), one-way, multi-city, strict arrival time, flexible dates, large group bookings, solo/business/family travel, weekend getaways, budget constraints, special occasions (e.g., birthdays) |
| **Hotel** | `<CHECK_IN_DATE_x>`, `<CHECK_OUT_DATE_x>`, `<CITY_x>`, `<CITY_x_NEIGHBORHOOD_y>`, `<STAR_x>`, `<RATING_x>`, `<PRICE_x>`, `<NUM_ROOMS_x>`, `<NUM_TRAVELERS_x>` | Multi-city trips, multiple hotels in one city, specific neighborhoods, special occasions, last-minute reservations, business vs. leisure, extended stays, view/accessibility/pet-friendly requests, family and group accommodations |
| **Attraction** | `<ATTRACTION_TYPE_x>`, `<CITY_x>`, `<STATE_x>`, `<CITY_x_NEIGHBORHOOD_y>` | Multi-type attraction queries, across cities/states, neighborhood-specific exploration, curiosity about state-level offerings. Assistant must ask open-ended questions and only mention placeholders if user provides them. |
| **Restaurant** | `<RESTAURANT_RATING_x>`, `<RESTAURANT_PRICE_x>`, `<RESTAURANT_CUISINE_x>`, `<CITY_x>`, `<CITY_x_NEIGHBORHOOD_y>` | Multi-city dining, cuisine preferences, dietary restrictions (e.g., nut-free, vegan, halal), budget vs. premium, special occasion dining, rating-focused or casual vs. upscale preferences |

Table 41: Domain-specific customization for template generation. Each domain builds on the shared prompt instructions with unique placeholders and required conversation scenarios.

# I  System Prompt Design for Multi-Domain Template Generation

This section outlines the system prompt instructions used for generating multi-turn conversations that span multiple domains. These prompts are tailored to create realistic, nonlinear interactions involving combinations of Flights, Hotels, Attractions, and Restaurants.

## I.1  Unified Prompt Instructions (Shared Across Multi-Domain Prompts)

- **Conversation Format:**

- Each template includes at least 8–10 dialogue turns (4–5 exchanges).
- Each template begins with an `assistant` turn and ends with a `user` turn.
- Turns alternate between `assistant:` and `user:`.

- **Placeholder Handling:**
    - Use only predefined placeholders (e.g., `<CITY_1>`, `<DEPARTURE_DATE_1>`).
    - Numbering restarts from 1 in every template.
    - Assistant must never mention a placeholder until the user provides it.

- **Assistant Behavior:**
    - Vary personality (formal, informal), verbosity, and strategies (direct vs. open-ended questions).
    - Must not assume user intent or combine domain-specific questions unless user initiates.
    - Must correct misunderstandings and adapt to user behavior.

- **User Behavior:**
    - Include users who are indecisive, impatient, verbose, overly specific, or vague.
    - Users may change their mind, skip steps, or backtrack in conversation.

- **Conversation Flow:**
    - Avoid rigid order (e.g., flight → hotel → attraction).
    - Encourage nonlinear scenarios: jumping between domains, multiple questions at once.

- **Language and Style:**
    - Vary greetings, transitions, sentence structures, and terminology across templates.
    - Avoid repeated phrasings.

- **Output Requirements:**
    - Format is strict JSON: `"template_1"` through `"template_30"`.
    - Each value is a newline-separated string of alternating assistant/user dialogue.
    - No preamble, postamble, or markdown allowed.

### I.2 Multi-Domain Scenario Requirements

| Domains | Placeholders Used | Required Scenarios |
|---|---|---|
| **Flight + Hotel** | `<CITY_x>`, `<CLASS_x>`, `<DEPARTURE_DATE_x>`, `<CHECK_IN_DATE_x>`, `<CHECK_OUT_DATE_x>`, `<NUM_TRAVELERS_x>` | Trip planning involving both air travel and accommodation. Includes round-trip and one-way flights, hotel stays across one or more cities, budget and luxury travelers, group vs. solo travel, flexible dates, specific preferences like non-stop flights or hotels with amenities (e.g., pool, gym, pet-friendly). |
| **Flight + Hotel + Attraction** | All flight, hotel, and attraction placeholders (e.g., `<CITY_x>`, `<CHECK_IN_DATE_x>`, `<ATTRACTION_TYPE_x>`) | User planning a full trip involving flights, hotel stays, and sightseeing. Must include: round-trip and one-way flights, multi-city stays, hotels near attractions, flexible schedules, user-provided attraction types or specific attraction names (only when linking to hotel proximity), and special constraints (e.g., group travel, business trips, anniversaries). Assistant must never use placeholder names before user introduces them. Includes amenity discussions and attraction-type grammar handling (e.g., cultural vs. `<ATTRACTION_TYPE_1>` attraction). |
| **Flight + Hotel + Restaurant** | All flight, hotel, and restaurant placeholders (e.g., `<RESTAURANT_CUISINE_x>`, `<RESTAURANT_PRICE_x>`) | End-to-end trip planning including flights, accommodation, and dining. Required scenarios include: dietary restrictions, cuisine-first or city-first planning, restaurant proximity to hotel, multiple hotels or restaurants in one city, round-trip and one-way flights, and amenities like pet-friendly or spa. Must feature diverse language, nonlinear flows, and varied user personas. Includes explicit handling of dietary needs (e.g., halal, nut-free, vegetarian), and placeholder logic as per prompt. |
| **Hotel + Attraction** | `<CITY_x>`, `<CHECK_IN_DATE_x>`, `<ATTRACTION_TYPE_x>`, `<CITY_x_NEIGHBORHOOD_y>` | User looking for accommodations and nearby attractions. Includes: specific neighborhoods, family-friendly vs. solo travel, themed vacations, extended stays, proximity to cultural or outdoor attractions, and cases where attraction type drives hotel location. |
| **Hotel + Restaurant** | `<RESTAURANT_CUISINE_x>`, `<HOTEL_PRICE_x>`, `<CITY_x_NEIGHBORHOOD_y>` | Combines hotel booking with dining preferences. Includes cuisine-specific searches, budget dining vs. fine dining, dietary needs (gluten-free, halal), special events (anniversaries), restaurant proximity to hotel, and group size considerations. Templates vary in planning flow—some start with hotel, others with restaurants. |

Table 42: Multi-domain combinations and scenario requirements. Each configuration builds on the unified instructions with additional placeholder logic and domain-specific complexity.

## J   Example System Prompt: Flight Domain

```
Your task is to generate diverse conversation templates for a flight finder chatbot.

Each template should capture realistic, UNIQUE conversations between a user and an
     assistant about flight searches.

Placeholders to Use:
- Departure city: <CITY_x>
- Destination arrival city: <CITY_x>
- Flight class: <CLASS_x>
- Departure date: <DEPARTURE_DATE_x>
- Departure time: <DEPARTURE_TIME_x>
- Destination arrival date: <ARRIVAL_DATE_x>
- Destination arrival time: <ARRIVAL_TIME_x>
- Airline name: <AIRLINE_x>
- Price: <PRICE_x>
- Airport's name: <AIRPORT_x>
- Number of travelers: <NUM_TRAVELERS_x>

Key Requirements:
1. Placeholder Numbering:
   - Use integers for ''x'' (e.g., <CITY_1>, <CITY_2>).
   - Reset numbering to 1 for each new template.

2. Conversation Format & Structure:
   - 6-8 turns minimum per template (3-4 exchanges).
   - Format as alternating assistant: and user: lines.
   - Each template must start with an assistant turn and end with a user turn.
   - Vary conversation lengths-some should be shorter, some longer.

3. Assistant Behavior Diversity:
   - Use distinctly different assistant personalities (formal, casual, verbose,
        concise).
   - Vary how information is requested (direct questions vs. open-ended).
   - Include templates where the assistant corrects misunderstandings.
   - NOTE: Assistant's first turn must be generic and not assume user intent (e.g.,
        ''How can I help you?").
   - NOTE: The assistant must not say it has found options for the user.

4. User Behavior Diversity:
   - Create dramatically different user types (impatient, polite, verbose, terse).
   - Include users who provide minimal information (requiring follow-ups).
   - Include users who provide too much information.
   - Include complex, specific requests.
   - Include indecisive users who change their minds.

5. Conversation Flow Diversity:
   - Avoid standard ''greeting -> dates -> preferences -> options'' structure.
   - Include nonlinear conversations (user circles back to earlier topics).
   - Include users asking multiple questions at once.
   - Include conversations skipping obvious steps.
   - Vary order in which info is elicited.

6. Required Scenarios:
   - Multi-city trips
   - Round trip
   - One-way trip
   - Flexible travel dates
   - Users with strong preferences (airline, class, airport, etc.)
   - Non-stop flights only
   - Short layovers are okay
   - Rigid arrival time
   - Special occasions (e.g., anniversaries)
```

```
                - Large group bookings
                - Last-minute reservations
                - Business travel
                - Family vacations
                - Solo travelers
                - Budget constraints
                - Weekend getaways

        7. Round Trip Requirement:
                - Roundtrip flights must include 2 departure dates.

        8. Multiple Travelers:
                - Always include number of travelers via placeholder in multi-person scenarios.

        9. Linguistic Diversity:
                - Avoid repeating same greetings across templates.
                - Vary language used for dates, preferences, and constraints.
                - Use different phrasings for similar ideas.
                - Unique transitions between topics.
                - Varied sentence structures and response styles.

        Important: Each template must be fundamentally different in structure, flow,
                language, and scenario.
```

## K    Example System Prompt: Flight-Hotel-Restaurant Domain

```
Your task is to generate diverse conversation templates for a travel assistant
        chatbot.

Each template should be a realistic and UNIQUE multi-turn interaction between a
        user and an assistant that includes:
- Flight search
- Hotel booking
- Restaurant recommendations or reservations

Placeholders to Use:
- Flight: <CITY_x>, <CLASS_x>, <DEPARTURE_DATE_x>, <DEPARTURE_TIME_x>,
        <ARRIVAL_DATE_x>, <ARRIVAL_TIME_x>, <AIRLINE_x>, <PRICE_x>, <AIRPORT_x>,
        <NUM_TRAVELERS_x>
- Hotel: <HOTEL_x>, <CHECKIN_DATE_x>, <CHECKOUT_DATE_x>, <ROOM_TYPE_x>,
        <NUM_GUESTS_x>, <PRICE_x>
- Restaurant: <RESTAURANT_x>, <CUISINE_x>, <RESERVATION_TIME_x>,
        <RESERVATION_DATE_x>, <PRICE_RANGE_x>, <LOCATION_x>, <NUM_PEOPLE_x>

Key Requirements:
1. Conversation must include all 3 components: flight, hotel, and restaurant.

2. Placeholder Numbering:
        - Use integers for ''x" (e.g., <CITY_1>, <CITY_2>).
        - Reset numbering to 1 for each new template.

3. Conversation Format:
        - At least 8 turns (4 full exchanges).
        - Alternating assistant: and user: lines.
        - Assistant always starts the conversation and user always ends it.
        - Vary length and structure across templates.

4. Assistant Behavior Diversity:
        - Include different tones: friendly, formal, efficient, humorous, inquisitive,
            etc.
        - Vary assistant strategies (confirming info early vs. late, etc.).
        - Assistant cannot confirm bookings or give specific results.
        - Must clarify ambiguous or missing information.
```

```
5. User Behavior Diversity:
   - Mix of confident and indecisive users.
   - Include users who change minds mid-conversation.
   - Include users who ask multiple things at once.
   - Include low-information and high-information users.

6. Conversation Flow Diversity:
   - Vary the sequence (flight -> hotel -> restaurant, or restaurant -> hotel ->
       flight, etc.).
   - Include users jumping between topics or circling back.
   - Include unexpected questions or constraints from users.

7. Required Scenarios:
   - Honeymoon trip
   - Last-minute business travel
   - Budget backpacking vacation
   - Solo food tourism
   - Family summer vacation
   - Couple anniversary trip
   - Conference trip with team
   - Weekend getaway
   - Luxury experience
   - Specific date events (concerts, sports games, etc.)

8. Linguistic Diversity:
   - Avoid repeating phrasing and structures across templates.
   - Vary tone and style significantly.
   - Mix concise and elaborate dialogues.

Important: Every template must feel distinct from all others in tone, language,
    structure, and scenario.
```

## L   Example System Prompt: Plan Generation Prompt

```
You are an expert AI travel planner and your responsibility is to generate Python
    code using APIs or Tools.
```

## M   Example User Prompt: Plan Generation Prompt

```
Your task is to generate a Python code based on a conversation between the user and
    the assistant, where the last turn is from the user.
The code typically involves calling one or more tools (functions) to help the user
    in planning their travel request.
In the Python code, you need to use the following tools:
# TOOL CONFIG
<TOOL_CONFIG>
# INSTRUCTIONS
- Track content: Maintain the conversation state across turns and use all known
    information from earlier in the conversation.
- As soon as the mandatory parameters (non-optional parameters) are all provided
    (refer to TOOL CONFIG to find mandatory parameters for each tool), generate
    the appropriate plan using Python code.
- Do NOT modify entity values under any circumstances. Use them exactly as they
    appear in the conversation while populating attributes in the function during
    code generation.
For example, if the city is "new york" (lowercase), do not convert it to "New York"
    or "NYC".
- Do not fill optional parameters unless they are explicitly provided in the
    conversation.
- When generating seek_information, only mention mandatory parameters (non-optional
    parameters) that are missing. Never ask for optional parameters using
    seek_information. Refer to TOOL CONFIG to figure out what the mandatory
    parameters (non optional parameters) are and check CONVERSATION to know what
    parameters have been provided by the user.
For example, "seek_information('<explain what mandatory parameters (non-optional
    parameters) are missing and must be gathered by the assistant>')"
- Only generate the code for the domain which the customer has mentioned in the
    conversation. For example, if user mentioned only about attractions, don't
    generate the code with restaurants search. Only if the user mentioned
    searching for restaurant anywhere in the conversation, then only search for
    restaurants.
- If a tool result from a previous turn is still valid and relevant, use
    get_results_from_cache(key="<cache_key>") to retrieve it. Use the cache
    summary to determine the most appropriate key to select from. If you have many
    keys in the cache for the same domain. Use the one which would be most
    relevant.
- If you generate a tool call and its result could be reused later, save it with
    save_to_cache("<key>",value). Ensure the cache key is unique and avoid naming
    collision with previously stored cache key name
- If a result has already been stored in the cache for a conversation and no new
    result needs to be generated, do not regenerate the code. Instead, return the
    code as "print("No planning needed")"
# OUTPUT FORMAT
- You need to generate the reasoning and the python code. The reasoning should
    clearly explain the process, steps and the reason behind the python plan that
    is going to be generated
The reasoning should be within the <REASONING> </REASONING> tags and the python
    code should be within the   tags. Note while generating the
    python code, never have any markdown tags.
# EXAMPLES
<FEW_SHOT_EXAMPLES>
# CONVERSATION
<CONVERSATION>
# CACHE
<CACHE_FOR_CONVERSATION>
Given the provided conversation and cache summary, generate a Python code for the
    last user turn.
```

## N   Template: Plan Generation for Fine-Tuning

You are an expert AI travel planner and your responsibility is to generate Python
    code using APIs or Tools.
Your task is to generate a Python code based on a conversation between the user and
    the assistant, where the last turn is from the user.
The code typically involves calling one or more tools (functions) to help the user
    in planning their travel request.
In the Python code, you need to use the following tools:
# TOOL CONFIG
<TOOL_CONFIG>
# INSTRUCTIONS
- Track content: Maintain the conversation state across turns and use all known
    information from earlier in the conversation.
- As soon as the mandatory parameters (non-optional parameters) are all provided
    (refer to TOOL CONFIG to find mandatory parameters for each tool), generate
    the appropriate plan using Python code.
- Do NOT modify entity values under any circumstances. Use them exactly as they
    appear in the conversation while populating attributes in the function during
    code generation.
    For example, if the city is "new york" (lowercase), do not convert it to "New
        York" or "NYC".
- Do not fill optional parameters unless they are explicitly provided in the
    conversation.
- When generating seek_information, only mention mandatory parameters (non-optional
    parameters) that are missing. Never ask for optional parameters using
    seek_information. Refer to TOOL CONFIG to figure out what the mandatory
    parameters (non optional parameters) are and check CONVERSATION to know what
    parameters have been provided by the user.
    For example, "seek_information('<explain what mandatory parameters
        (non-optional parameters) are missing and must be gathered by the
        assistant>')"
- Only generate the code for the domain which the customer has mentioned in the
    conversation. For example, if user mentioned only about attractions, don't
    generate the code with restaurants search. Only if the user mentioned
    searching for restaurant anywhere in the conversation, then only search for
    restaurants.
- If a tool result from a previous turn is still valid and relevant, use
    get_results_from_cache(key="<cache_key>") to retrieve it. Use the cache
    summary to determine the most appropriate key to select from. If you have many
    keys in the cache for the same domain. Use the one which would be most
    relevant.
- If you generate a tool call and its result could be reused later, save it with
    save_to_cache("<key>",value). Ensure the cache key is unique and avoid naming
    collision with previously stored cache key name
- If a result has already been stored in the cache for a conversation and no new
    result needs to be generated, do not regenerate the code. Instead, return the
    code as "print("No planning needed")"
# OUTPUT FORMAT
- You need to generate only the python code. The python code should be within the
      tags. Note while generating the python code, never have any
    markdown tags.
# CONVERSATION
<CONVERSATION>
# CACHE
<CACHE_FOR_CONVERSATION>
Given the provided conversation and cache summary, generate a Python code for the
    last user turn.

