# OpenReview forum: "T1: A Tool-Oriented Conversational Dataset for Multi-Turn Agentic Planning"
_NeurIPS.cc/2025/Datasets_and_Benchmarks_Track — NeurIPS 2025 Datasets and Benchmarks Track poster_

### Official Review · Reviewer_Mk2Q · 2025-07-01

**Rating:** 5
**Confidence:** 3

**Summary:**

This paper introduces T1, a novel tool-oriented, multi-turn, multi-domain conversational dataset specifically designed to evaluate LLM-based agents' capabilities in planning and executing complex tool-use workflows. The T1 dataset includes 13.5k human-verified dialogues involving 14 tools across nine domains, simulating realistic assistant behavior that requires inter-tool dependencies, context-aware planning, and memory-based caching. To complement the dataset, the authors also present T1-AGENT, a benchmark agent equipped with a caching mechanism and code generation abilities. Experimental results using both fine-tuned and pre-trained LLMs demonstrate that while domain adaptation significantly improves tool-use accuracy, multi-domain reasoning and generalization remain open challenges.

**Additional Feedback:**

1. It would be valuable to analyze the failure cases qualitatively, especially in the multi-domain setting—e.g., what types of tool dependencies most often lead to incorrect planning?

2. Incorporating interleaved natural feedback (e.g., user clarification or tool failure handling) could simulate more realistic human-agent interactions.

**Dataset Code Accessibility:**

Yes

**Dataset Code Comments:**

The benchmark codebase appears to be complete and well-maintained, although I have not executed it myself.

**Ethical Considerations:**

No, there are no or only very minor ethics concerns

**Final Justification:**

The additional explanations and supporting evidence have improved my understanding of the work and its contributions. Based on this, I have decided to raise my score accordingly.

**Limitations Weaknesses:**

1. The experiments are restricted to open-source LLMs (LLaMA 3.1 8B and 70B), which may not reflect the upper bound of current agentic planning capabilities, particularly in commercial applications.
2. The paper does not provide error bars or variance measures across runs, which limits the assessment of model stability and robustness across dialogue samples.
3. The current domains (e.g., flights, hotels, restaurants, attractions) are mostly focused on travel-related scenarios. The generalizability to other domains like finance, healthcare, or software engineering remains unclear.

**Strengths Contributions:**

1. The T1 dataset addresses an important gap by simulating complex, tool-dependent, multi-turn conversations with cross-domain reasoning, going beyond prior benchmarks that focus on single-turn or isolated tool usage.
2. All dialogues are carefully constructed through a three-step templating and lexicalization pipeline, validated by experienced annotators with strong programming backgrounds, ensuring code correctness and coherence.
3. The paper defines a rich set of evaluation metrics (tool call accuracy, parameter matching F1, execution success, information seeking similarity, etc.), enabling detailed analysis of agent performance across settings.

---

> ### Author Rebuttal · Authors · 2025-07-30
>
> Thank you very much for taking the time to review our paper and for your valuable recommendations. We sincerely appreciate your insightful comments and have addressed the weaknesses and limitations you pointed out in the sections below.
>
> ## Response to Limitation 1: ##
>
> We already reported **Phi-4-reasoning-plus** and **S1.1-32B** in our supplementary materials to show the performance of reasoning models on our full datasets. We did not provide the results in the main section due to space limitation. In order to address Limitation 1, we ran additional experiments on **2** proprietary LLMs **(OpenAI o4-mini and Gemini 2.5 Pro)**. In total, this experiment includes **7** LLMs (5 open-weight, 2 proprietary) using our datasets. Running these LLMs on the full dataset would cost over **$5,000**, so we created a random subset, representing **4%** of the data, that evenly covers all **9** domains and includes a diverse sample of templates within each domain. This allows for efficient and cost-effective benchmarking. All **7** models were evaluated on this subset. We plan to open source both the full and the subset datasets.
>
> Below are the results of all models on the subset dataset. Proprietary models consistently outperform all open-weight models that we tested (including the reasoning models **S1.1 32B** and **Phi-4-reasoning-plus**). Among open-weight models, the SFT model achieves the strongest performance across most metrics.
>
> **Overall Results using LLMs:**
>
> | **Domain** | **Tool Call Acc.** | **Tool Call Prec.** | **Tool Call Rec.** | **Tool Call F1** | **Parameter Matching Acc.** | **Parameter Matching Prec.** | **Parameter Matching Rec.** | **Parameter Matching F1** | **Code Exec. Rate Acc.** | **Information Seeking SacreBLEU** | **Information Seeking BERTScore** | **Cache Summary EM** |
> | :--- | :---: | :---: | :---: | :---: | :---: | :---: | :---: | :---: | :---: | :---: | :---: | :---: |
> | **`Open Weight`** | | | | | | | | | | | | |
> | Phi-4-reasoning-plus | 54.27 | 75.45 | 64.45 | 69.14 | 35.98 | 60.20 | 48.84 | 51.84 | 86.45 | 27.26 | 80.91 | 40.91 |
> | S1.1 32B | 71.15 | 79.04 | **87.41** | 82.72 | 54.60 | 62.83 | **79.78** | 70.01 | 60.65 | 17.65 | 79.72 | 52.75 |
> | LLAMA-3.3 70B Instruct | 67.02 | 76.20 | 83.03 | 79.14 | 53.65 | 62.11 | 76.74 | 67.54 | **91.44** | 31.38 | 82.87 | 57.17 |
> | LLAMA-3.1 8B Instruct | 35.87 | 44.63 | 64.33 | 52.25 | 19.45 | 24.71 | 49.91 | 32.25 | 42.35 | 27.12 | 81.41 | 29.45 |
> | LLAMA-3.1 8B SFT | **77.48** | **86.87** | **87.41** | **87.07** | **60.68** | **73.16** | 78.35 | **75.39** | 83.77 | **42.39** | **85.77** | **63.79** |
> | **`Proprietary`** | | | | | | | | | | | | |
> | Gemini 2.5 Pro | **89.32** | **93.49** | **95.18** | **94.28** | 73.68 | 82.49 | 87.04 | 84.63 | **94.46** | 7.11 | 74.44 | **76.11** |
> | OpenAI o4-mini | 86.47 | 91.88 | 93.44 | 92.59 | **77.31** | **86.91** | **87.32** | **87.05** | 89.00 | **22.93** | **81.52** | 72.43 |
>
> ## Response to Limitation 2: ##
> To address your comments on Limitation 2, we assessed the models’ output quality and stability across runs with different seeds. We tested **LLAMA-3.3 70B Instruct** (open-source) and **Gemini 2.5 Pro** (proprietary) using seeds **64**, **100**, and **200**. We found that code generation outputs were highly consistent with minimal variation. Due to cost and time constraints, experiments were limited to these two models on a **4%** sample of the test set spanning all 9 domains.
>
> ### Model Performance Metrics
>
> | **Model** | **Tool Call Parameter Matching Acc.** | **Tool Call Parameter Matching Prec.** | **Tool Call Parameter Matching Rec.** | **Tool Call Parameter Matching F1** | **Code Exec. Rate Acc.** | **Code Exec. Rate Prec.** | **Code Exec. Rate Rec.** | **Code Exec. Rate F1** | **Information Seeking Acc.** | **Information Seeking SacreBLEU** | **Information Seeking BERTScore** | **Information Seeking EM** |
> | :--- | :---: | :---: | :---: | :---: | :---: | :---: | :---: | :---: | :---: | :---: | :---: | :---: |
> | Gemini 2.5 Pro | $88.93 \pm 0.35$ | $93.26 \pm 0.20$ | $94.92 \pm 0.26$ | $94.04 \pm 0.21$ | $73.07 \pm 0.54$ | $81.97 \pm 0.45$ | $86.69 \pm 0.33$ | $84.19 \pm 0.39$ | $94.23 \pm 0.24$ | $7.56 \pm 0.40$ | $74.54 \pm 0.13$ | $75.44 \pm 0.77$ |
> | LLAMA-3.3 70B Instruct | $66.98 \pm 0.12$ | $76.51 \pm 0.38$ | $82.77 \pm 0.42$ | $79.15 \pm 0.04$ | $53.40 \pm 0.22$ | $61.91 \pm 0.39$ | $76.81 \pm 0.06$ | $67.48 \pm 0.20$ | $91.65 \pm 0.64$ | $31.04 \pm 0.30$ | $82.73 \pm 0.12$ | $57.55 \pm 0.33$ |
>
> ## Response to Limitation 3: ##
> Our T1 dataset was built using a **domain-agnostic framework** and we selected **4** travel-related domains that are inherently complex, interdependent, and involve multi-turn dialogues. We believe that even through the travel domain we can portray enough nuances to demonstrate the intricacy and value of the framework. These domains were combined to create **9** distinct categories, including **single** and **multi-domain** scenarios. The chosen domains are highly realistic and reflect use cases found in real-world applications such as searching for a hotel near a particular tourist attraction or searching for flights and hotels for a multi-city trip. That said, our primary objective is not to develop a general-purpose agent, and generalizability to unrelated domains falls outside the scope of this work. We plan to explore additional domains in future work and will revise the paper to provide more details on the rationale behind our domain selection, as suggested.
>
> ## Response to Additional Feedback 1: ##
>
> To address the reviewer’s comment about analyzing the tool dependencies in failure cases, we conducted a quantitative analysis to identify failure modes in code generation across various domains and to assess which tool combinations most commonly lead to errors. The experiment was done with all **7** models on our **4%** subset dataset. Our findings indicate that reasoning-based models consistently struggle in scenarios requiring the combined use of search_attractions and filter_attractions, accounting for approximately **6%-7%** of all tool call failures. This pattern typically occurs when a user both requests new attraction recommendations and simultaneously wishes to refine previously retrieved results stored in the cache. In contrast, non-reasoning models had a different error distribution without any generalizable patterns.
>
> ## Response to Additional Feedback 2: ##
> This is a great idea and we will consider this in our future work.

---

> > ### Author Response · Authors · 2025-08-04
> > **Friendly Reminder to Reply to our Rebuttal**
> >
> > Dear Reviewer,
> >
> > I hope you are doing well. Please review our rebuttal. We have addressed all of your comments and suggestions. We only have 2 more days until the deadline so your response would be greatly appreciated.

---

> > > ### Comment · Reviewer_Mk2Q · 2025-08-06
> > >
> > > Thank you for the authors' response.
> > >
> > > Due to my oversight, I initially missed the relevant information provided in the Supplementary Material. The authors have indeed included additional experimental results for different models, which is appreciated and encouraged. However, this remains insufficient. The paper should include performance comparisons across different model sizes within the same model series.
> > >
> > > In addition, regarding the randomness of LLMs, the authors mentioned that the code generation outputs were highly consistent with minimal variation. Could the authors provide further explanation for this observation?
> > >
> > > Moreover, the paper still sets its scenario within the domain of travel planning, which limits its novelty given the current state of benchmarks. That said, the work focuses primarily on multi-turn dialogue, which does contribute a degree of originality.

---

### Official Review · Reviewer_rTbs · 2025-07-01

**Rating:** 4
**Confidence:** 4

**Summary:**

This paper introduces T1, a tool-augmented, multi-domain, multi-turn conversational dataset aimed at evaluating and improving large language models (LLMs) in complex planning scenarios involving interdependent API/tool calls. The authors also present T1-AGENT, a baseline system used to demonstrate the benchmark's effectiveness.

**Additional Feedback:**

In Figure 1, it should be "When do you plan to fly to San Jose?"

**Dataset Code Accessibility:**

Partly

**Dataset Code Comments:**

I don't find any readme file of the dataset in the released repo.

**Ethical Considerations:**

No, there are no or only very minor ethics concerns

**Final Justification:**

I appreciate the authors' response that addressed most of my concern. I will raise my score

**Limitations Weaknesses:**

1. It is not clear to me how the multi-turn aspect is evaluated. Based on my understanding, multi-turn here means model will interact with user multiple times, but it seems like during evaluation, there is no real user involved, which confused me. Take the example of Figure 1, during evaluation, the assistant may not be able to ask the exact question shown in Figure 1, then how should the evaluation proceed? If the dialogue is fixed and the evaluated model is just to output the tool call, then it's not a real multi-turn scenario. To me, multi-turn should be user can give feedback or follow-up question regarding llm's results.
2. Experiment is quite limited. The evaluation only involves 3 models and all are llama3.1 series, while the training experiment only involves llama 3.1. This limited experiment is quite inconclusive.

**Strengths Contributions:**

1. The paper addresses a clear gap in the direction of LLM tool using: the lack of dataset for multi-turn tool using LLM.
2. The T1 dataset covers nine distinct domains (including both single- and multi-domain tasks), supporting rich evaluation of multi-tool coordination, dependency resolution, and memory management.
3. The introduction of T1-AGENT demonstrates how the dataset can be used to train and evaluate real systems.

---

> ### Author Rebuttal · Authors · 2025-07-30
>
> Thank you very much for taking the time to review our paper and for your valuable recommendations. We sincerely appreciate your insightful comments and have addressed the weaknesses and limitations you pointed out in the sections below.
>
> ## Response to Limitation 1: ##
>
> The **T1 dataset** is designed to evaluate the planning capabilities of agentic models within multi-turn conversations, rather than simulating real-time, interactive agent-to-user dialogues. Our focus is on assessing a model's ability to generate appropriate tool calls based on the provided conversational context. This approach prioritizes a consistent and deterministic evaluation setting for Large Language Models (LLMs), minimizing the variability inherent in live user interactions. The T1 dataset was created similar to established multi-turn dialogue datasets like MultiWOZ [1] and SVD [2] which is considered a standard way to create multi-turn dialogue datasets. **These datasets are widely used in conversational agent research and offer a more consistent and deterministic evaluation setting for LLM's as they eliminate the variability of real-time user input.**
>
> In the **T1 dataset**, each multi-turn dialogue is structured as a user-agent role-play. The user acts as a customer seeking information on travel services (e.g., flights, hotels, restaurants, attractions) and asks follow-up questions to refine their requests. The agent, in the role of a travel assistant, is tasked with responding by identifying and planning the execution of the relevant tools to fulfill the user's request, based on the ongoing conversation.
> **T1 Agent**, our **planning agent**, is responsible for **generating a plan** similar to [3], but in a complex dialogue setting. This process involves using the **entire conversation history**, **cached data**, and **available tools** to determine the correct sequence and parameters for tool calls. So it's this generated plan that we then **evaluate**. We are not evaluating an end-to-end conversational agent that drives the entire dialogue or engages in free-form conversation. Instead, T1 Agent, assesses the efficacy of the planning agent in accurately interpreting nuanced user intents across turns and generating executable code for a predefined set of tools, including managing inter-tool dependencies and leveraging caching mechanisms. This setup allows for a rigorous evaluation of an LLM's capacity for complex planning and reasoning in context-rich, goal-oriented scenarios, which is a significant challenge in modern multi-turn conversational systems.
>
> [1] Budzianowski, P., Wen, T. H., Tseng, B. H., Casanueva, I., Ultes, S., Ramadan, O., & Gašić, M. (2018). Multiwoz--a large-scale multi-domain wizard-of-oz dataset for task-oriented dialogue modelling. arXiv preprint arXiv:1810.00278.
>
> [2] Rastogi, A., Zang, X., Sunkara, S., Gupta, R., & Khaitan, P. (2020, April). Towards scalable multi-domain conversational agents: The schema-guided dialogue dataset. In Proceedings of the AAAI conference on artificial intelligence (Vol. 34, No. 05, pp. 8689-8696).
>
> [3] T. Schick, J. Dwivedi-Yu, R. Dessì, R. Raileanu, M. Lomeli, E. Hambro, L. Zettlemoyer,
> N. Cancedda, and T. Scialom. Toolformer: Language models can teach themselves to use tools.
> Advances in Neural Information Processing Systems, 36:68539–68551, 2023
>
> ## Response to Limitation 2: ##
>
> In our main paper, we reported results from **Llama-3.3 70B Instruct** in addition to Llama-3.1 models. Additionally, we also reported **Phi-4-reasoning-plus** and **S1.1-32B** in our supplementary materials to show the performance of reasoning models on our full datasets. We did not provide the results in the main section due to space limitation. In order to address Limitation 2, we ran additional experiments on **2** proprietary LLMs **(OpenAI o4-mini and Gemini 2.5 Pro)**. In total, this experiment includes **7** LLMs (5 open-weight, 2 proprietary) using our datasets. Running these LLMs on the full dataset would cost over **$5,000**, so we created a random subset, representing **4%** of the data, that evenly covers all **9** domains and includes a diverse sample of templates within each domain. This allows for efficient and cost-effective benchmarking. All **7** models were evaluated on this subset. We plan to open source both the full and the subset datasets.
>
> Below are the results of all models on the subset dataset. Proprietary models consistently outperform all open-weight models that we tested (including the reasoning models **S1.1 32B** and **Phi-4-reasoning-plus**). Among open-weight models, the SFT model achieves the strongest performance across most metrics.
>
> **Overall Results using LLMs:**
>
> | **Domain** | **Tool Call Acc.** | **Tool Call Prec.** | **Tool Call Rec.** | **Tool Call F1** | **Parameter Matching Acc.** | **Parameter Matching Prec.** | **Parameter Matching Rec.** | **Parameter Matching F1** | **Code Exec. Rate Acc.** | **Information Seeking SacreBLEU** | **Information Seeking BERTScore** | **Cache Summary EM** |
> | :--- | :---: | :---: | :---: | :---: | :---: | :---: | :---: | :---: | :---: | :---: | :---: | :---: |
> | **`Open Weight`** | | | | | | | | | | | | |
> | Phi-4-reasoning-plus | 54.27 | 75.45 | 64.45 | 69.14 | 35.98 | 60.20 | 48.84 | 51.84 | 86.45 | 27.26 | 80.91 | 40.91 |
> | S1.1 32B | 71.15 | 79.04 | **87.41** | 82.72 | 54.60 | 62.83 | **79.78** | 70.01 | 60.65 | 17.65 | 79.72 | 52.75 |
> | LLAMA-3.3 70B Instruct | 67.02 | 76.20 | 83.03 | 79.14 | 53.65 | 62.11 | 76.74 | 67.54 | **91.44** | 31.38 | 82.87 | 57.17 |
> | LLAMA-3.1 8B Instruct | 35.87 | 44.63 | 64.33 | 52.25 | 19.45 | 24.71 | 49.91 | 32.25 | 42.35 | 27.12 | 81.41 | 29.45 |
> | LLAMA-3.1 8B SFT | **77.48** | **86.87** | **87.41** | **87.07** | **60.68** | **73.16** | 78.35 | **75.39** | 83.77 | **42.39** | **85.77** | **63.79** |
> | **`Proprietary`** | | | | | | | | | | | | |
> | Gemini 2.5 Pro | **89.32** | **93.49** | **95.18** | **94.28** | 73.68 | 82.49 | 87.04 | 84.63 | **94.46** | 7.11 | 74.44 | **76.11** |
> | OpenAI o4-mini | 86.47 | 91.88 | 93.44 | 92.59 | **77.31** | **86.91** | **87.32** | **87.05** | 89.00 | **22.93** | **81.52** | 72.43 |
>
>
> ## Response to Additional Feedback: ##
>
> Thanks for catching this typo. We will make sure to correct it in the final version.

---

> > ### Author Response · Authors · 2025-08-04
> > **Friendly Reminder to Reply to our Rebuttal**
> >
> > Dear Reviewer,
> >
> > I hope you are doing well. Please review our rebuttal. We have addressed all of your comments and suggestions. We only have 2 more days until the deadline so your response would be greatly appreciated.

---

> > > ### Author Response · Authors · 2025-08-07
> > > **Reminder to Reply to our Rebuttal - 1 more day left (after extension)**
> > >
> > > Dear Reviewer,
> > >
> > > I hope you are doing well. Please review our rebuttal. We have addressed all of your comments and suggestions. We only have 1 more day left (after extension) until the deadline so your response would be greatly appreciated.

---

> > ### Comment · Reviewer_rTbs · 2025-08-08
> >
> > I appreciate the authors' response that addressed most of my concern. I will raise my score

---

> > > ### Author Response · Authors · 2025-08-08
> > > **Thank you for the acknowledgement and raising the score**
> > >
> > > Thank you for taking the time to review our responses and for raising the score. We truly appreciate it.

---

### Official Review · Reviewer_KqFq · 2025-07-06

**Rating:** 4
**Confidence:** 3

**Summary:**

This paper introduces T1, a conversational dataset for evaluating the multi-turn planning capabilities of LLM-based agents on traveling domain. A key contribution is an integrated caching mechanism designed to improve efficiency by reusing tool call results. The authors also present T1-AGENT, a baseline agent, and provide an analysis of its performance.

**Additional Feedback:**

See Weaknesses

**Dataset Code Accessibility:**

Yes

**Ethical Considerations:**

No, there are no or only very minor ethics concerns

**Final Justification:**

Most concerns addressed.

**Limitations Weaknesses:**

1.  **Limited Generalizability:** The dataset is highly focused on the travel domain. This specialization raises questions about the generalizability of agents trained on T1. The authors should discuss the potential limitations in applying these models to other domains, or consider refine the statement to clearly mention that this work mainly focus on the traveling domain.
2.  **Data Generation and Diversity:** The use of templates to generate dialogue risks a lack of linguistic diversity. It is recommended that the authors quantify the linguistic variety within the dataset and perhaps contrast their template-based method with an alternative where placeholders are filled more dynamically by a generative model, such as GPT-4o, which might yield more natural and varied language.
3.  **Incomplete Evaluation:** The evaluation could be significantly strengthened.
    *   **Model Benchmarking:** The assessment should be expanded to include a wider range of contemporary models, including leading open-source (e.g., Qwen, DeepSeek) and proprietary systems (e.g., GPT-4o, Claude 3 Sonnet, Gemini 2.5 Pro), to provide a more comprehensive view of the current state-of-the-art on this task.
    *   **Error Analysis:** The paper would benefit from a detailed error analysis. It is crucial to understand *why* agents fail. The analysis should categorize failures, such as syntax errors in generated code, incorrect tool parameters, or instruction-following errors in complex multi-turn flows.
4.  **Lack of Empirical Data on Caching:** While the caching mechanism is a core contribution, its benefits are not empirically substantiated. The authors should provide quantitative data, including the average frequency of cache use, the success/failure rate of cache retrieval, and the resulting token savings. An ablation study comparing performance with and without caching would be critical to demonstrating its impact on final task success.
5.  **Clarity on Artifact Release:** The authors mention in the contributions and appendix their intent to release the dataset and code. This is commendable. To maximize visibility and impact, this intention should be stated more prominently, perhaps even in the abstract, to ensure the community is immediately aware of the resource's availability.

**Strengths Contributions:**

1.  **Multi-Turn Dialogue Structure:** The dataset's focus on multi-turn conversations.
2.  **Efficient Context via Caching:** The introduction of a caching mechanism is a practical approach to simplifying context management and reducing computational overhead. This feature encourages the development of more efficient agents that can intelligently reuse previous results rather than repeating expensive tool invocations.

---

> ### Author Rebuttal · Authors · 2025-07-30
>
> Thank you very much for taking the time to review our paper and for your valuable recommendations. We sincerely appreciate your insightful comments and have addressed the weaknesses and limitations you pointed out in the sections below.
>
> ## Response to Limited Generalizability: ##
> Our T1 dataset was built using a domain-agnostic framework and we selected **4** travel-related domains that are inherently complex, interdependent, and involve multi-turn dialogues. We believe that even through the travel domain we can portray enough nuances to demonstrate the intricacy and value of the framework. These domains were combined to create **9** distinct categories, including **single** and **multi-domain** scenarios. The chosen domains are highly realistic and reflect use cases found in real-world applications such as searching for a hotel near a particular tourist attraction or searching for flights and hotels for a multi-city trip. That said, our primary objective is not to develop a general-purpose agent, and generalizability to unrelated domains falls outside the scope of this work. We plan to explore additional domains in future work and will revise the paper to provide more details on the rationale behind our domain selection, as suggested.
>
> ## Response to Data Generation and Diversity: ##
> For our paper, we created **60** diverse templates per domain, covering both single and multi-domain conversations to ensure lexical variability and to support large-scale dataset generation. We appreciate your comments about quantifying linguistic diversity, therefore, we additionally computed the Moving Average Type to Token Ratio (**MATTR**) which is a measure of lexical diversity that quantifies vocabulary variability using a **50-word** window. Our dataset averages a **MATTR** score above **0.7**, indicating **high lexical diversity**.
> Please find the domain-specific **MATTR** scores below.
>
> | Category                  | Value |
> | ------------------------- | ----- |
> | Attraction                | 0.68  |
> | Flight                    | 0.74  |
> | Hotel                     | 0.76  |
> | Restaurant                | 0.76  |
> | Flight-Hotel-Attraction   | 0.72  |
> | Flight-Hotel-Restaurant   | 0.71  |
> | Flight-Hotel              | 0.73  |
> | Hotel-Attraction          | 0.73  |
> | Hotel-Restaurant          | 0.75  |
> As part of our **template generation** process, we used a decoder model, specifically **Llama-3.3 70B Instruct**, to paraphrase and refine the templates for improved naturalness. We conducted a manual evaluation with **human annotators** to assess the quality of the generated multi-turn templates and found that they are both fluent and natural sounding. While other models such as **GPT-4o** could also serve as decoder models for generating dialogues with specific entities, exploring their use is beyond the scope of this work and is left for future research.
>
> ## Response to Model Benchmarking: ##
> We already reported **Phi-4-reasoning-plus** and **S1.1-32B** in our supplementary materials to show the performance of reasoning models on our full datasets. We did not provide the results in the main section due to space limitation. In order to address the Model Benchmarking limitation, we ran additional experiments on **2** proprietary LLMs **(OpenAI o4-mini and Gemini 2.5 Pro)**. In total, this experiment includes **7** LLMs (5 open-weight, 2 proprietary) using our datasets. Running these LLMs on the full dataset would cost over **$5,000**, so we created a random subset, representing **4%** of the data, that evenly covers all **9** domains and includes a diverse sample of templates within each domain. This allows for efficient and cost-effective benchmarking. All **7** models were evaluated on this subset. We plan to open source both the full and the subset datasets.
>
> Below are the results of all models on the subset dataset. Proprietary models consistently outperform all open-weight models that we tested (including the reasoning models **S1.1 32B** and **Phi-4-reasoning-plus**). Among open-weight models, the SFT model achieves the strongest performance across most metrics.
>
> **Overall Results using LLMs:**
>
> | **Domain** | **Tool Call Acc.** | **Tool Call Prec.** | **Tool Call Rec.** | **Tool Call F1** | **Parameter Matching Acc.** | **Parameter Matching Prec.** | **Parameter Matching Rec.** | **Parameter Matching F1** | **Code Exec. Rate Acc.** | **Information Seeking SacreBLEU** | **Information Seeking BERTScore** | **Cache Summary EM** |
> | :--- | :---: | :---: | :---: | :---: | :---: | :---: | :---: | :---: | :---: | :---: | :---: | :---: |
> | **`Open Weight`** | | | | | | | | | | | | |
> | Phi-4-reasoning-plus | 54.27 | 75.45 | 64.45 | 69.14 | 35.98 | 60.20 | 48.84 | 51.84 | 86.45 | 27.26 | 80.91 | 40.91 |
> | S1.1 32B | 71.15 | 79.04 | **87.41** | 82.72 | 54.60 | 62.83 | **79.78** | 70.01 | 60.65 | 17.65 | 79.72 | 52.75 |
> | Llama-3.3 70B Instruct | 67.02 | 76.20 | 83.03 | 79.14 | 53.65 | 62.11 | 76.74 | 67.54 | **91.44** | 31.38 | 82.87 | 57.17 |
> | Llama-3.1 8B Instruct | 35.87 | 44.63 | 64.33 | 52.25 | 19.45 | 24.71 | 49.91 | 32.25 | 42.35 | 27.12 | 81.41 | 29.45 |
> | Llama-3.1 8B SFT | **77.48** | **86.87** | **87.41** | **87.07** | **60.68** | **73.16** | 78.35 | **75.39** | 83.77 | **42.39** | **85.77** | **63.79** |
> | **`Proprietary`** | | | | | | | | | | | | |
> | Gemini 2.5 Pro | **89.32** | **93.49** | **95.18** | **94.28** | 73.68 | 82.49 | 87.04 | 84.63 | **94.46** | 7.11 | 74.44 | **76.11** |
> | OpenAI o4-mini | 86.47 | 91.88 | 93.44 | 92.59 | **77.31** | **86.91** | **87.32** | **87.05** | 89.00 | **22.93** | **81.52** | 72.43 |
>
>
>
> ## Response to Error Analysis: ##
> We fully agree that analyzing agent failures is important for understanding model behavior. In the **supplementary materials section E**, we have already included an initial error analysis, categorizing failures into four types: **Validation Error** (invalid tool arguments), **Variable Not Defined**, **Index Out of Range** and **Other** (miscellaneous code issues including **syntax errors**). Our evaluation suggests that models struggle with effective cache utilization and generating correct code, particularly in complex, multi-turn scenarios. Task-specific fine-tuning, however, significantly enhances performance, indicating that a lack of domain adaptation contributes to these errors.
>
>
> We hope this analysis helps provide clarity on the types of errors observed during inference.
>
>
> ## Response to Lack of Empirical Data on Caching: ##
> We thank the reviewer for recognizing caching as one of the core contributions.
> To address this, we have computed additional metrics. Here are each metric's definitions and implementation:
>
>
> - **Success Rate (Retrieve Cache):** This metric measures how often the model successfully retrieves the correct information from a previously saved cache when it calls **get_results_from_cache**. It reflects the model's ability to reuse prior knowledge or data effectively during a multi-turn conversation.
>
> - **Avg. Tool Call:** This represents the **average number of tool calls made per user turn**. It helps indicate how frequently the model relies on external tools (e.g., API calls, function executions) in a typical turn.
>
> - **Avg. Save Cache:** This is the **average number of save_to_cache calls per user turn**, showing how often the model chooses to save information to cache. It reflects the model’s tendency to store intermediate results or key facts for future retrieval.
>
> - **Avg. Retrieve Cache:** This is the **average number of get_results_from_cache calls per user turn**, showing how often the model tries to retrieve previously saved information during the conversation.
>
> The metrics for all models are as following:
> [**The values closer to the reference number are considered better**]
>
>
> | Model             | Success Rate (Retrieve Cache) | Avg. Tool Call | Avg. Save Cache | Avg. Retrieve Cache |
> |-------------------|-------------------------------|----------------|------------------|----------------------|
> | **Reference**         | 100.00  | 2.0330 | 0.70550          | 0.4321383            |
> | **`Open Weight`** | | | | | | | | | | | | |
> | Phi-4-reasoning-plus             | 58.58                         | 1.5679         | 0.49245          | 0.3570190            |
> | S1.1 32B             | 55.41                         | 1.6592         | 0.58690          | 0.3534780            |
> | Llama-3.3 70B Instruct    | 61.51 | 2.1822 | 0.80290          | 0.3847380            |
> | Llama-3.1 8B Instruct     | 22.57| 1.8760 | 0.60580          | 0.2380218            |
> | Llama-3.1 8B Instruct SFT  | **88.85**| 2.1095 | 0.77227          | 0.4668910            |
> | **`Proprietary`** | | | | | | | | | | | | |
> | Gemini-2.5 Pro           | 87.58 | 2.0530         | 0.72979          | 0.4004211            |
> | GPT-o4-mini                | 87.31   | **2.0323**         | **0.69830**          | **0.4382163**            |
>
> Thank you for raising your concern about **token saving**. Just to clarify, the primary motivation behind implementing the cache is not to reduce the number of tokens, but to **avoid redundant tool calls** and **reducing system latency**. To the best of our knowledge, existing literature primarily focuses on making fresh tool calls at every turn. In contrast, our approach leverages previously made tool calls, so if a relevant call was made in an earlier user turn and its result can be reused or filtered as needed, we retrieve the output from the cache instead of making extra identical calls.
>
> ## Response to Clarity on Artifact Release: ##
> We thank the reviewer for bringing this up. We would be happy to add more details and clear intentions about releasing the dataset and code in the final paper.

---

> > ### Author Response · Authors · 2025-08-04
> > **Friendly Reminder to Reply to our Rebuttal**
> >
> > Dear Reviewer,
> >
> > I hope you are doing well. Please review our rebuttal. We have addressed all of your comments and suggestions. We only have 2 more days until the deadline so your response would be greatly appreciated.

---

> > > ### Author Response · Authors · 2025-08-07
> > > **Reminder to Reply to our Rebuttal - 1 more day left (after extension)**
> > >
> > > Dear Reviewer,
> > >
> > > I hope you are doing well. Please review our rebuttal. We have addressed all of your comments and suggestions. We only have 1 more day left (after extension) until the deadline so your response would be greatly appreciated.

---

> > > > ### Author Response · Authors · 2025-08-08
> > > > **Final Reminder to Reply to our Rebuttal**
> > > >
> > > > Dear Reviewer,
> > > >
> > > > I hope you are doing well. We have addressed all of your comments and suggestions. Today is the final day and your response would be greatly appreciated.

---

> > ### Comment · Reviewer_KqFq · 2025-08-09
> > **Thanks**
> >
> > Thanks for your rebuttal. It solves most of my concerns; therefore I am raising the score to 4.

---

### Author Response · Authors · 2025-08-07
**Summary of our Answers (1/2)**

Dear AC and Reviewers,

We extend our gratitude to all reviewers for your **valuable feedback** and **raising the scores**. We have addressed all questions and concerns raised. Thank you for your time and consideration and we would be happy to answer any questions before the deadline.

The following summary outlines the content we included in our rebuttal to assist the AC and all reviewers in following the discussion. Additionally, we have addressed each reviewer’s questions individually.
- **Benchmarking on 9 more models and Adding model scaling:** We run state-of-the-art LLMs across nine domains presented a significant challenge as requested by Reviewers KqFq, rTbs, and Mk2Q with nine additional models, including the S1.1 (1.5B, 3B, 7B) and Llama 3 series (3.1 70B Instruct), as well as proprietary models like GPT4.1, GPT o3, and Gemini 2.5 Pro. Analysis of the results revealed key findings on model scaling and performance discrepancies between open-weight and proprietary models. Smaller S1.1 models (1.5B and 3B) struggled with code generation, leading to zero SacreBLEU and BERTScore values due to incorrect and unparsable code. Performance and metrics improved with increasing model sizes for both open-weight and proprietary models, with open-weight models showing enhancements starting from the 7B model, a trend also observed in the GPT4.1 series. All results were generated using a dataset of 540 multi-turn dialogue samples across 9 domains to optimize experimental costs, as a full dataset run would have exceeded $5000, which was beyond our resource constraints. We will add the following results and findings on the final paper.

| **Model** | **Tool Call Acc.** | **Tool Call Prec.** | **Tool Call Rec.** | **Tool Call F1** | **Parameter Matching Acc.** | **Parameter Matching Prec.** | **Parameter Matching Rec.** | **Parameter Matching F1** | **Code Exec. Rate Acc.** | **Information Seeking SacreBLEU** | **Information Seeking BERTScore** | **Cache Summary EM** |
|:---|:---:|:---:|:---:|:---:|:---:|:---:|:---:|:---:|:---:|:---:|:---:|:---:|
| **Open Weight** | | | | | | | | | | | | |
| Llama-3.1 8B SFT | 77.48 | 86.87 | 87.41 | 87.07 | 60.68 | 73.16 | 78.35 | 75.39 | 83.77 | **42.39** | **85.77** | 63.79 |
| Llama-3.1 70B Instruct | 72.28 | 80.06 | 87.49 | 83.49 | 59.16 | 68.79 | 80.61 | 73.59 | 77.62 | 32.46 | 83.20 | 53.94 |
| Llama-3.3 70B Instruct | 67.02 | 76.20 | 83.03 | 79.14 | 53.65 | 62.11 | 76.74 | 67.54 | 91.44 | 31.38 | 82.87 | 57.17 |
| Llama-3.1 8B Instruct | 35.87 | 44.63 | 64.33 | 52.25 | 19.45 | 24.71 | 49.91 | 32.25 | 42.35 | 27.12 | 81.41 | 29.45 |
| Phi-4-reasoning-plus | 54.27 | 75.45 | 64.45 | 69.14 | 35.98 | 60.20 | 48.84 | 51.84 | 86.45 | 27.26 | 80.91 | 40.91 |
| S1.1 32B | 71.15 | 79.04 | 87.41 | 82.72 | 54.60 | 62.83 | 79.78 | 70.01 | 60.65 | 17.65 | 79.72 | 52.75 |
| S1.1 7B | 30.45 | 46.03 | 52.66 | 40.09 | 19.39 | 32.48 | 37.98 | 27.78 | 5.37 | 12.74 | 69.32 | 27.18 |
| S1.1 3B | 28.05 | 37.36 | 48.05 | 41.28 | 13.99 | 16.95 | 39.34 | 23.16 | 5.98 | 0.00 | 0.00 | 26.78 |
| S1.1 1.5B | 5.93 | 5.93 | 11.11 | 7.73 | 0.00 | 0.00 | 0.00 | 0.00 | 3.95 | 0.00 | 0.00 | 26.98 |
| **Proprietary** | | | | | | | | | | | | |
| Gemini 2.5 Pro | **89.32** | 93.49 | **95.18** | **94.28** | 73.68 | 82.49 | 87.04 | 84.63 | **94.46** | 7.11 | 74.44 | **76.11** |
| OpenAI o3 | 85.41 | 89.60 | 94.76 | 91.91 | 75.18 | 83.77 | **87.88** | 85.64 | 93.51 | 22.52 | 82.26 | 73.53 |
| OpenAI o4-mini | 86.47 | 91.88 | 93.44 | 92.59 | **77.31** | **86.91** | 87.32 | **87.05** | 89.00 | 22.93 | 81.52 | 72.43 |
| GPT 4.1 | 86.00 | 92.02 | 92.68 | 92.32 | 75.01 | 84.49 | 86.79 | 85.53 | 91.20 | 29.37 | 83.54 | 73.80 |
| GPT 4.1-mini | 83.99 | **93.77** | 88.64 | 91.09 | 72.96 | 83.92 | 84.70 | 84.20 | 87.29 | 20.83 | 81.12 | 67.27 |
| GPT 4.1-nano | 63.14 | 75.38 | 78.13 | 76.46 | 39.60 | 52.63 | 61.31 | 56.26 | 77.90 | 19.79 | 79.16 | 40.65 |

- **Lexical Diversity:** Additionally, Reviewer KqFq also asked about the diversity of the text within our templates and we therefore computed the MATTR score from the templates in all 9 domains.
Empirical Caching Data: For the cache, , Reviewer KqFq asked to provide some empirical data on caching and therefore included metrics for the success rate of the retrieved cache, the average tool calls made per user turn, the average number of times an item is saved to the cache during a user turn and the average number of times an item is retrieved from the cache.
Tool Dependency Failure Analysis: We also conducted some analysis on tool dependency failures and which scenarios had the highest number of failures within tool calls as requested by Reviewer Mk2Q.

---

> ### Author Response · Authors · 2025-08-07
> **Summary of our Answers (2/2)**
>
> - **Model Stability and Error Bars:** Additionally, Reviewer Mk2Q asked us for error bars and to give any evidence that we observed the output quality and stability of the model through several runs. We ended up computing metrics for Llama 3.3 70B Instruct and Gemini 2.5 Pro across three seed values and observed in the end that the model output quality was relatively stable even across different seed values. The standard deviation of our models are relatively small, less than 1 point.
>
> | **Model** | **Tool Call Parameter Matching Acc.** | **Tool Call Parameter Matching Prec.** | **Tool Call Parameter Matching Rec.** | **Tool Call Parameter Matching F1** | **Code Exec. Rate Acc.** | **Code Exec. Rate Prec.** | **Code Exec. Rate Rec.** | **Code Exec. Rate F1** | **Information Seeking Acc.** | **Information Seeking SacreBLEU** | **Information Seeking BERTScore** | **Information Seeking EM** |
> | :--- | :---: | :---: | :---: | :---: | :---: | :---: | :---: | :---: | :---: | :---: | :---: | :---: |
> | Gemini 2.5 Pro | $88.93 \pm 0.35$ | $93.26 \pm 0.20$ | $94.92 \pm 0.26$ | $94.04 \pm 0.21$ | $73.07 \pm 0.54$ | $81.97 \pm 0.45$ | $86.69 \pm 0.33$ | $84.19 \pm 0.39$ | $94.23 \pm 0.24$ | $7.56 \pm 0.40$ | $74.54 \pm 0.13$ | $75.44 \pm 0.77$ |
> | LLAMA-3.3 70B Instruct | $66.98 \pm 0.12$ | $76.51 \pm 0.38$ | $82.77 \pm 0.42$ | $79.15 \pm 0.04$ | $53.40 \pm 0.22$ | $61.91 \pm 0.39$ | $76.81 \pm 0.06$ | $67.48 \pm 0.20$ | $91.65 \pm 0.64$ | $31.04 \pm 0.30$ | $82.73 \pm 0.12$ | $57.55 \pm 0.33$ |
>
> - **Dataset Purpose and Generalizability Clarification:** Reviewer KqFq -  We also had to clarify on the generalizability of our dataset as well as the main purpose. There was some belief that our dataset generated synthetic conversations between the assistant and user but that is not the purpose of our dataset. Instead, the idea is that the T1 dataset would be used as a method of evaluating the ability of the model in generating the correct planned code using the provided tools.

---

### Note · Authors · 2025-08-12

Hi AC and SACs,

We extend our gratitude to **all reviewers for your valuable feedback and raising the scores**. We have addressed all questions and concerns raised. Additionally, all improvements will be added to the final paper. Thank you for your time and consideration.

The details are as follows:
- **Benchmarking on 9 more models and adding model scaling:** In response to requests from reviewers KqFq, rTbs, and Mk2Q, we extended our evaluation across the nine domains presented in the paper by testing nine additional state-of-the-art language models. These include the S1.1 models (1.5B, 3B, 7B), the Llama 3 series (including the 70B Instruct model), and proprietary models such as GPT-4.1, o3, and Gemini 2.5 Pro. Our analysis uncovered key insights into model scaling and the performance gap between open-weight and proprietary systems. Notably, the smaller S1.1 models (1.5B and 3B) performed poorly on code generation tasks, resulting in zero SacreBLEU and BERTScore metrics due to producing incorrect code. As model size increased, both open-weight and proprietary models demonstrated improved performance, with open-weight models showing marked gains starting at the 7B scale—a pattern that mirrored trends observed in proprietary models like GPT-4.1.
- **Lexical Diversity**: In response to Reviewer KqFq's question regarding the textual diversity of our templates, we computed the MATTR (Moving-Average Type-Token Ratio) scores across all nine domains.
- **Empirical Caching Data**: In response to Reviewer KqFq’s request for empirical data on caching, we included metrics such as the cache retrieval success rate, the average number of tool calls per user turn, the average number of items saved to the cache per user turn, and the average number of cache retrievals per turn.
- **Tool Dependency Failure Analysis:** As requested by Reviewer Mk2Q, we also analyzed tool dependency failures, identifying the scenarios with the highest failure rates during tool calls.

---

### Decision · Program_Chairs · 2025-09-18

**Decision:**

Accept (poster)

**Comment:**

After rebuttal, reviewers are in general convinced by the contribution of the work. While the work focused on travel-related tasks (which can be narrow in nature), it provides in-depth data (including flights, restaurants, hotels, attractions), templates to generate multi-round conversations around these entities, templates on generating symbolic plans given the conversations, and relatively thorough experiments on top of the dataset. Overall it is a nice contribution to the community.